# Modeling Multi-Task Model Merging as Adaptive Projective Gradient Descent

Yongxian Wei [1]  Anke Tang [2]  Li Shen [3]  Zixuan Hu [4]  Chun Yuan [1]  Xiaochun Cao [3]

## Abstract

Merging multiple expert models offers a promising approach for performing multi-task learning without accessing their original data. Existing methods attempt to alleviate task conflicts by sparsifying task vectors or promoting orthogonality among them. However, they overlook the fundamental target of model merging: the merged model performs as closely as possible to task-specific models on respective tasks. We find these methods inevitably discard task-specific information that, while causing conflicts, is crucial for performance. Based on our findings, we frame model merging as a constrained optimization problem (*i.e.*, minimizing the gap between the merged model and individual models, subject to the constraint of retaining shared knowledge) and solve it via adaptive projective gradient descent. Specifically, we align the merged model with individual models by decomposing and reconstituting the loss function, alleviating conflicts through *data-free* optimization of task vectors. To retain shared knowledge, we optimize this objective by projecting gradients within a *shared subspace* spanning all tasks. Moreover, we view merging coefficients as adaptive learning rates and propose a task-aware, training-free strategy. Experiments show that our plug-and-play approach consistently outperforms previous methods, achieving state-of-the-art results across diverse architectures and tasks in both vision and NLP domains. Our code is available here.

## 1. Introduction

Fine-tuning pre-trained foundational models to address downstream tasks has become an effective paradigm (Muqeeth et al., 2024). However, the independent deployment of multiple fine-tuned models increases storage costs. While traditional multi-task learning (MTL) can mitigate these issues, they typically require concurrent training across multiple task-specific datasets, which incurs significant training overhead and potential privacy risks (Wei et al., 2024a). Consequently, there is a growing interest in merging multiple expert models into a unified model without accessing their original data (Yang et al., 2024a; Huang et al., 2024). Model merging is performed directly at the parameter level and maintains only one final model during inference. In recent years, numerous pre-trained and fine-tuned checkpoints have been released on open-source communities like GitHub or Hugging Face, making it easy to obtain expert models from diverse domains. These rich model repositories underscore the value of model merging.

One popular approach, Task Arithmetic (TA) (Ilharco et al., 2023), combines task vectors through arithmetic operations for model merging. A major challenge is addressing conflicts that emerge when multiple task-specific models coexist within a single model. Ties-Merging (Yadav et al., 2023) proposes pruning redundant parameters, resolving sign conflicts, and merging sparse models, while AdaMerging (Yang et al., 2024c) applies test-time adaptation techniques to adjust merging coefficients in the weight space. Most recently, AWD (Xiong et al., 2024) finds that orthogonality among task vectors is key to model merging and introduces adaptive weight disentanglement to improve orthogonality. However, these methods overlook the fundamental requirement of model merging: ensuring the merged model performs comparably to task-specific models on their respective tasks.

Revisiting multi-task model merging, we make the following findings: (i) As the number of tasks increases, existing methods inevitably discard task-specific information that, while causing conflicts, is crucial for performance. (ii) Task vectors are inherently close to orthogonal. Further promoting orthogonality results in the loss of shared knowledge, especially when tasks are similar. (iii) Merging coefficients share a similarity with learning rates in MTL, considering task vectors actually represent accumulated gradients.

Based on our rethinking, we frame model merging as a constrained optimization problem (*i.e.*, minimizing the gap between the merged model and individual models, subject

[1]Tsinghua University [2]Wuhan University [3]Shenzhen Campus of Sun Yat-sen University [4]Nanyang Technological University. Correspondence to: Li Shen <mathshenli@gmail.com>, Chun Yuan <yuanc@sz.tsinghua.edu.cn>.

*Proceedings of the 42$^{nd}$ International Conference on Machine Learning*, Vancouver, Canada. PMLR 267, 2025. Copyright 2025 by the author(s).

to the constraint of retaining shared knowledge) and solve it via an a**d**aptive pr**o**jective **g**radient d**e**scent (DOGE) method. Specifically, we measure the gap between the merged model and individual models in task-specific losses, and decompose it into a data-free objective using the first-order Taylor expansion. To alleviate conflicts, we introduce a modification vector $\Delta$ (*i.e.*, redundant parameters) to each task vector. This data-free objective aims to achieve optimal average performance across multiple tasks by optimizing $\Delta$. For the modification vector, task vectors still compete to minimize the loss on their own tasks. Therefore, we construct a shared subspace based on all task vectors and optimize the problem within this subspace. The gradient of $\Delta$ can be divided into two components: one projected onto the shared subspace and the other orthogonal to it. We only take gradient steps in the direction orthogonal to the shared space, effectively constraining task vector optimization. As the former represents movements of parameters within the shared subspace, and the latter maintains shared knowledge while minimizing the gap for each task. Moreover, we determine task-aware, training-free merging coefficients based on the norm of task vectors to mitigate the dominance of any single task's gradient influence.

We conduct experiments on diverse vision and NLP tasks, including classification and generation, using various fully fine-tuned and LoRA fine-tuned architectures. Our plug-and-play approach achieves up to 11.6% gains over TA and 5.8% over AdaMerging. Simple task-aware $\lambda$ provides a 2.8% performance boost. Furthermore, experiments on unseen tasks and out-of-distribution test sets demonstrate its generalization and robustness. Extensive ablation studies clarify the mechanisms of each component.

In summary, our main contributions are three-fold:

- We rethink model merging from a multi-task learning perspective, and model it as a constrained optimization problem that aims to mitigate task conflicts while retaining shared knowledge.

- We propose adaptive projective gradient descent, a novel approach that optimizes a data-free objective within a shared subspace and incorporates task-aware, training-free merging coefficients.

- We conduct comprehensive experiments and discussions; our empirical results demonstrate a significant improvement over previous methods.

## 2. Related Work

**Model merging.** Model merging (Crisostomi et al., 2024; Wang et al., 2024b; Daheim et al., 2024; Chen et al., 2024; Maldonado et al., 2024) eliminates the need for raw training data or expensive computations. It operates directly at

the parameter level and consolidates multiple models into a single final model for inference. Existing model merging methods are categorized into two paradigms: pre-merging and during-merging (Yang et al., 2024a). Pre-merging methods aim to create favorable conditions for merging, such as using linearized fine-tuning to achieve weight disentanglement (Ortiz-Jimenez et al., 2023; Tang et al., 2024c).

During-merging methods focus on developing techniques to merge given models and can be broadly categorized into data-free and test-time adaptation (TTA) approaches. TTA methods assume access to unlabeled test datasets and are often considered a form of transductive learning. For example, AdaMerging (Yang et al., 2024c) learns merging coefficients by minimizing entropy as a surrogate loss on test data, while Representation Surgery (Yang et al., 2024b) calibrates biases and aligns the merged model's representations with those of the original task-specific models. In contrast, our approach designs a fully data-free objective to resolve task conflicts without relying on test data.

Data-free methods depend solely on the pre-trained and fine-tuned model weights for merging (Choi et al., 2024). Ties-Merging (Yadav et al., 2023) prunes redundant parameters by magnitude, resolves sign conflicts, and merges sparse models. Concrete Merging (Tang et al., 2023) adopts a meta-learning framework to learn a concrete mask that suppresses conflicting parameters. MAP (Li et al., 2025a) examines task vector magnitudes and leverages a second-order Taylor expansion to approximate loss-based metrics, providing a formal bound on the remainder term and using linear regression to estimate the Hessian.

Calculating the loss gap has been reflected in some studies: MetaGPT (Zhou et al., 2024) formally defines the loss difference and derives a closed-form solution for the merging coefficient $\lambda$. TATR (Sun et al., 2025) introduces the concept of knowledge conflict between tasks by modeling the loss gap as the product of gradients and task vectors. Other relevant works explore merging within subspaces. TSV (Gargiulo et al., 2024) aggregates task vectors using low-rank approximation and whitening to minimize interference, while KnOTS (Stoica et al., 2025) aligns representation spaces between LoRA models via SVD to enable compatible merging. These approaches, like ours, recognize the inherent low-rank structure of parameter updates and perform merging within subspaces. We focus on optimizing task vectors via gradient descent while constraining it within a shared subspace to retain shared knowledge.

**Multi-task learning.** Existing MTL research addresses the issue of negative transfer (Jiang et al., 2023) from two principal perspectives: architecture and optimization. From the architectural perspective, negative transfer is mitigated through strategies like modularization (Lu et al., 2024), spar-

sification (Sun et al., 2020), or soft sharing of the backbone. From an optimization perspective, it is widely recognized that tasks sharing similar underlying structures can benefit from being trained together. Gradient alignment methods (Yu et al., 2020; Shi et al., 2022) emphasize maintaining consistency in gradient directions or signs to resolve conflicts, which projects one task's gradient onto the normal plane of another task's gradient to reduce forgetting (Saha et al., 2021). Our approach enhances multi-task performance by aligning the merged model with each individual model and utilizing adaptive merging coefficients.

## 3. Revisit Model Merging

In this section, we first introduce the problem setup and notations for model merging, followed by a rethinking of model merging from a multi-task learning perspective.

### 3.1. Preliminary

We begin with a pre-trained model $f$, parameterized by $\boldsymbol{\theta}_0$, which has been trained on a large-scale dataset. This model is paired with a collection of $n$ downstream tasks, denoted as $\{\mathcal{D}_i\}_{i=1}^n$. Subsequently, the pre-trained model $f$ is fine-tuned individually for each downstream task $\mathcal{D}_i$, resulting in a series of fine-tuned models, each with its unique parameters $\boldsymbol{\theta}_i$. To isolate task-specific information, we define the task vector as $\boldsymbol{\tau}_i = \boldsymbol{\theta}_i - \boldsymbol{\theta}_0$, a concept introduced by Ilharco et al. (2023). The set of these task vectors is represented as $\{\boldsymbol{\tau}_i\}_{i=1}^n$, enabling a focused analysis of the task-specific characteristics. Model merging aims to compose a multi-task model $\boldsymbol{\theta}^*$ to approximate the optimal solution:

$$\boldsymbol{\theta}_{opt} \approx \boldsymbol{\theta}^* = \mathcal{A}(\boldsymbol{\theta}_0, \boldsymbol{\tau}_1, \cdots, \boldsymbol{\tau}_n). \quad (1)$$

Here, $\mathcal{A}$ represents an arbitrary merging algorithm. For instance, in Task Arithmetic, $\boldsymbol{\theta}^* = \boldsymbol{\theta}_0 + \lambda \sum_{i=1}^n \boldsymbol{\tau}_i$.

### 3.2. Rethinking Model Merging for MTL

**How to resolve conflicts among parameters?** Resolving conflicts among tasks is a key challenge in model merging. Unlike MTL, which can mitigate conflicts during training with access to original data, model merging operates entirely in the parameter space. Existing methods mainly address conflicts by sparsely adjusting parameters, either by dropping conflicting parameters based on signs (Yadav et al., 2023) or importance scores (Du et al., 2024). Other methods promote orthogonality among task vectors, either by fine-tuning models in the tangent space (Ortiz-Jimenez et al., 2023) or directly optimizing task vectors (Xiong et al., 2024). While these methods alleviate conflicts, they inevitably discard task-specific information that contributes to conflicts, resulting in performance degradation. However, they overlook the fundamental target of model merging:

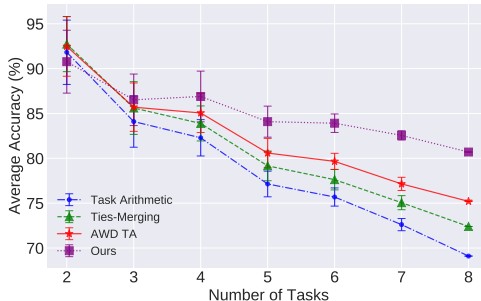

*Figure 1.* The effect of task numbers on average accuracy for ViT-B/32, with error bars representing the 95% confidence interval. As the number of tasks increases, negative transfer becomes more pronounced. Although our method initially performs lower than other methods, its performance decreases more slowly, demonstrating superior robustness when handling a larger number of tasks.

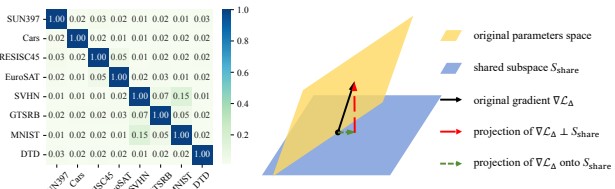

*Figure 2.* (a) Cosine similarity matrices of task vectors for ViT-B/32. (b) A schematic representation of the subspace spanned by the task representations, depicted as a two-dimensional plane.

the merged model performs as closely as possible to task-specific models on respective tasks. As shown in Fig. 1, increasing the task numbers leads to a continuous performance decline across methods. This is because more tasks result in increased negative transfer, causing the discard of valuable conflict-related task-specific knowledge. Therefore, we propose explicitly modeling the gap between the merged model $\boldsymbol{\theta}^*$ and individual models $\boldsymbol{\theta}_i$. This transforms conflict resolution into an optimization problem that can be solved using gradient descent.

**Is shared knowledge retained?** In addition to resolving conflicts, MTL should also encourage shared representations—a crucial aspect overlooked by existing methods. Experiments reveal that sparsely retaining parameters across tasks results in disjoint parameter dimensions, causing a loss of shared knowledge. Fig. 2(a) shows the cosine similarity between task vectors, which is *inherently small*, consistent with the theorem that high-dimensional vectors tend to be almost orthogonal (Vershynin, 2018). This explains the success of methods like TA. However, further increasing orthogonality to mitigate conflicts can exacerbate shared knowledge loss. Parameters between similar tasks are shareable (*e.g.*, applying the MNIST task vector improves accuracy on SVHN). Therefore, we propose constructing a

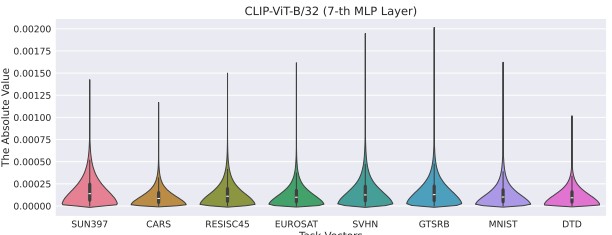

*Figure 3.* An illustration of element magnitudes in the task vector, inspired by (Shen et al., 2024). Best viewed when zoomed in.

shared subspace $S_{\text{share}}$ to preserve common representations (see Fig. 2(b)). This involves constraining task vector optimization to reduce updates along $S_{\text{share}}$.

**What is the role of $\lambda$?**    A critical observation is the importance of the merging coefficient $\lambda$. In methods like TA, a unified $\lambda$ is searched on the validation set. Ideally, $\lambda$ values should be task- and layer-specific. However, when dealing with a large number of tasks and layers, traditional methods such as grid search or combinatorial optimization search (Liu et al., 2020) become impractical. TTA methods require training $\lambda$ using unlabeled test data, which also presents limitations. A statistical analysis of task vector values reveals that tasks and layers exhibit different magnitudes (see Fig. 3). Modern adaptive optimizers (*e.g.*, Adam) dynamically adjust learning rates based on gradient history, which is often more effective than a global learning rate. These optimizers suppress parameters with large gradients and reward those with small gradients, smoothing gradient fluctuations. Similarly, task vectors $\boldsymbol{\tau}_i$ represent cumulative gradients for each task, and $\lambda$ can be viewed as a learning rate balancing gradients across multiple tasks.

## 4. Methodology

Based on above findings, we frame model merging as a constrained optimization problem (*i.e.*, minimizing the gap while the position in the subspace remains unchanged):

$$
\begin{aligned}
\min_{\boldsymbol{\theta}^*} \quad & \mathcal{L}(\Delta; \lambda_i, \boldsymbol{\tau}_i) := \sum_{i=1}^{n} \text{Gap}(\boldsymbol{\theta}^*, \boldsymbol{\theta}_i), \\
\text{s.t.} \quad & S_{\text{share}}(\boldsymbol{\theta}^*, \boldsymbol{\theta}_0 + \lambda \sum_{i=1}^{n} \boldsymbol{\tau}_i) = 0.
\end{aligned} \tag{2}
$$

Here, $\Delta$ is initialized as a zero tensor with the same shape as the task vector. The function $\text{Gap}(\cdot, \cdot)$ measures the distance between two sets of parameters, while $S_{\text{share}}(\cdot, \cdot)$ denotes the distance within the shared subspace. Then, we solve it via adaptive projective gradient descent:

$$
\Delta = \Delta - \mathbf{g}, \text{ where } \mathbf{g} = \text{Proj}_{\perp S_{\text{share}}}\big(\nabla_\Delta \mathcal{L}(\Delta; \lambda_i, \boldsymbol{\tau}_i)\big).
$$

It uses adaptive $\lambda_i$ for different tasks, projects the gradient orthogonal to $S_{\text{share}}$ to satisfy the constraint, and optimizes

the modification of $\boldsymbol{\theta}^*$ to minimize the loss.

In Sec. 4.1, we introduce an optimizable modification vector $\Delta$ using gradient descent to reduce the gap. In Sec. 4.2, we construct the shared subspace $S_{\text{share}}$ and project the objective into this subspace for optimization. Finally, in Sec. 4.3, we introduce the adaptive merging coefficient $\lambda_i$.

### 4.1. A Data-Free Objective

Considering the fundamental target that the merged model should perform comparably to its respective task-specific model for each task, we follow Zhou et al. (2024) to define the objective for resolving model merging as:

$$
\min \sum_{j=1}^{n} \left( \mathcal{L}_j(\boldsymbol{\theta}_0 + \lambda \sum_{i=1}^{n} \boldsymbol{\tau}_i) - \mathcal{L}_j(\boldsymbol{\theta}_0 + \boldsymbol{\tau}_j) \right)^2, \tag{3}
$$

where $\mathcal{L}_j(\boldsymbol{\theta})$ denotes the loss for task $j$ with model parameters $\boldsymbol{\theta}$. This objective requires that the merged model's performance on each task closely matches the performance achieved using only the corresponding task vector $\boldsymbol{\tau}_j$.

Multi-task conflicts often arise during model merging, as expert models encapsulate diverse and sometimes conflicting knowledge. Therefore, we introduce a modification vector $\Delta$ to each task vector, aiming to alleviate conflicts by optimizing $\Delta$. Previous work (Xiong et al., 2024) has shown that eliminating redundant components from task vectors can help reduce interference between tasks. In this context, $\Delta$ can be understood as the shared redundant portion of task vectors. However, directly optimizing Eq. (3) requires task-specific data to compute $\mathcal{L}_j$, which is unavailable as we only have access to model parameters. To overcome this limitation, we apply a Taylor expansion around the pre-trained model parameters $\boldsymbol{\theta}_0$ (Ortiz-Jimenez et al., 2023):

$$
\begin{aligned}
\min_{\Delta} & \sum_{j=1}^{n} \left( \mathcal{L}_j(\boldsymbol{\theta}_0 + \lambda \sum_{i=1}^{n}(\boldsymbol{\tau}_i + \Delta)) - \mathcal{L}_j(\boldsymbol{\theta}_0 + \boldsymbol{\tau}_j) \right)^2 \\
\approx \min_{\Delta} & \sum_{j=1}^{n} \left( \mathcal{L}_j(\boldsymbol{\theta}_0) + \langle \nabla_{\boldsymbol{\theta}} \mathcal{L}_j(\boldsymbol{\theta}_0), \lambda \sum_{i=1}^{n}(\boldsymbol{\tau}_i + \Delta) \rangle \right. \\
& \left. - \mathcal{L}_j(\boldsymbol{\theta}_0) - \langle \nabla_{\boldsymbol{\theta}} \mathcal{L}_j(\boldsymbol{\theta}_0), \boldsymbol{\tau}_j \rangle \right)^2 \\
= \min_{\Delta} & \sum_{j=1}^{n} \left( \langle \nabla_{\boldsymbol{\theta}} \mathcal{L}_j(\boldsymbol{\theta}_0), \lambda \sum_{i=1}^{n}(\boldsymbol{\tau}_i + \Delta) - \boldsymbol{\tau}_j \rangle \right)^2.
\end{aligned} \tag{4}
$$

Similarly, calculating the gradient $\nabla_{\boldsymbol{\theta}} \mathcal{L}_j(\boldsymbol{\theta}_0)$ of the pre-trained model for task $j$ requires access to data $\mathcal{D}_j$, which is typically unavailable. As an alternative, we approximate this gradient using the task vector $-\boldsymbol{\tau}_j$, since the task vector can be interpreted as an accumulation of gradients. Under the Neural Tangent Kernel assumption (*i.e.*, fine-tuning oc-

curs in a linear regime), $\nabla_{\boldsymbol{\theta}}\mathcal{L}_j(\boldsymbol{\theta}_0)$ can be estimated as $k\boldsymbol{\tau}_j$ with $k < 0$. Here, $\boldsymbol{\tau}_j = \boldsymbol{\theta}_T - \boldsymbol{\theta}_0 = -\sum_{t=1}^{T}\alpha_t\nabla_{\boldsymbol{\theta}_t}\mathcal{L}_j(\boldsymbol{\theta}_t)$, where $\alpha_t$ is the learning rate and $T$ is the number of update steps. Given that parameters remain near $\boldsymbol{\theta}_0$, we have $\nabla_{\boldsymbol{\theta}_t}\mathcal{L}_j(\boldsymbol{\theta}_t) = \nabla_{\boldsymbol{\theta}_0}\mathcal{L}_j(\boldsymbol{\theta}_0)$. Thus, we obtain $\nabla_{\boldsymbol{\theta}}\mathcal{L}_j(\boldsymbol{\theta}_0) = -\boldsymbol{\tau}_j/\sum_{t=1}^{T}\alpha_t$. Consequently, the data-free objective can be approximated as:

$$\min_{\Delta}\sum_{j=1}^{n}\left(\left\langle-\boldsymbol{\tau}_j, \lambda\sum_{i=1}^{n}(\boldsymbol{\tau}_i + \Delta) - \boldsymbol{\tau}_j\right\rangle\right)^2. \quad (5)$$

The set of task vectors $\{\boldsymbol{\tau}_i\}_{i=1}^{n}$ is known, and Eq. (5) represents a data-free objective that optimizes the modification vector $\Delta$ based on model parameters. This can be solved using optimizers such as gradient descent, enabling the merged model to achieve enhanced performance on specific tasks. Next, we illustrate how to perform optimization within a shared subspace through gradient projection.

### 4.2. Shared Subspace Optimization

Model merging promotes multi-tasking capabilities within a single model, which inevitably leads to parameter competition across tasks. For the modification vector $\Delta$, each task competes to minimize the loss of the merged model on its own task. Towards this end, we construct a shared subspace for all tasks to retain shared knowledge.

Let $S_j = span\{\boldsymbol{B}_j\}$ represent the subspace spanned by the task vector $\boldsymbol{\tau}_j$, where $\boldsymbol{B}_j = [\boldsymbol{u}_{j,1}, ..., \boldsymbol{u}_{j,k}]$ is the basis matrix for $S_j$, consisting of $k$ basis vectors extracted from task vector $\boldsymbol{\tau}_j$. For any matrix $\boldsymbol{A}$ with suitable dimensions, its projection onto subspace $S_j$ is defined as:

$$\mathrm{Proj}_{S_j}(\boldsymbol{A}) = \boldsymbol{B}_j(\boldsymbol{B}_j)^{\top}\boldsymbol{A}. \quad (6)$$

We utilize Singular Value Decomposition (SVD) to extract the rank-$k$ subspace for the task vector. Specifically, the first $k$ singular vectors from the left singular matrix are selected as $\boldsymbol{B}_j$, forming an orthogonal basis that efficiently captures the primary information within the task-specific $\boldsymbol{\tau}_j$. Once the subspaces for all tasks are established, they are combined into a shared subspace $S_{\mathrm{share}} = span\{[\boldsymbol{B}_1, ..., \boldsymbol{B}_n]\}$. However, $S_{\mathrm{share}}$ includes overlapping singular vectors, indicating redundant parameters in the weight space across tasks. Such overlaps challenge the orthogonality requirement of basis vectors and lead to inaccuracies during projection onto the shared subspace. To mitigate this, we perform another SVD on $S_{\mathrm{share}}$ to deduplicate it further, resulting in a refined $S_{\mathrm{share}}$ that effectively preserves shared knowledge.

Eq. (5) can be projected onto the shared subspace, which allows the gradient to be decomposed into two distinct components: (i) a component projected onto $S_{\mathrm{share}}$, which induces

---

**Algorithm 1:** Adaptive Projective Gradient Descent

**Input** : Pre-trained model $\boldsymbol{\theta}_0$; Fine-tuned models $\{\boldsymbol{\theta}_i\}_{i=1}^{n}$; Subspace basis size $k$; Global scaling factor $\eta$.

**Output** : Merged multi-task model $\boldsymbol{\theta}^*$.

// Task-Wise Preparation
**for** $i \leftarrow 1$ **to** $n$ **do**
  Compute task vector $\boldsymbol{\tau}_i \leftarrow \boldsymbol{\theta}_i - \boldsymbol{\theta}_0$
  Compute merging coefficients $\lambda_i^l \leftarrow \frac{\eta}{\|\boldsymbol{\tau}_i^l\|}$
  Perform SVD on $\boldsymbol{\tau}_i$: $\boldsymbol{\tau}_i = U_i\Sigma_i V_i^{\top}$
  $\boldsymbol{B}_i \leftarrow$ the first $k$ columns of $U_i$

// Construct the Shared Subspace
$S_{\mathrm{share}} \leftarrow$ the first $k$ columns of $U$ from $\mathrm{SVD}([\boldsymbol{B}_1, ..., \boldsymbol{B}_n])$

// Optimize $\Delta$ in the Subspace
**for** iteration $\leftarrow 1$ **to** $T$ **do**
  $\mathcal{L}(\Delta) \leftarrow \sum_{j=1}^{n}\left\langle-\boldsymbol{\tau}_j, \sum_{i=1}^{n}\lambda_i(\boldsymbol{\tau}_i + \Delta) - \boldsymbol{\tau}_j\right\rangle^2$
  $\nabla_{\Delta}\mathcal{L} \leftarrow \nabla_{\Delta}\mathcal{L} - \mathrm{Proj}_{S_{\mathrm{share}}}(\nabla_{\Delta}\mathcal{L})$
  Update $\Delta$ via gradient descent

**return** $\boldsymbol{\theta}^* \leftarrow \boldsymbol{\theta}_0 + \sum_{i=1}^{n}\lambda_i(\boldsymbol{\tau}_i + \Delta)$

---

parameter updates $\lambda\sum_{i=1}^{n}(\boldsymbol{\tau}_i + \Delta)$ within the shared subspace; (ii) the other component lies in the direction orthogonal to $S_{\mathrm{share}}$ when learning Eq. (5). Notably, this component optimizes $\Delta$ without altering the shared knowledge, while minimizing the gap for task $j$. Thus, before taking a gradient step, the new gradients $\nabla_{\Delta}\mathcal{L}$ are first projected onto $S_{\mathrm{share}}$. The projected components are then subtracted from the new gradient, leaving only the components orthogonal to $S_{\mathrm{share}}$. The updated gradients are calculated as:

$$\nabla_{\Delta}\mathcal{L} = \nabla_{\Delta}\mathcal{L} - \mathrm{Proj}_{S_{\mathrm{share}}}(\nabla_{\Delta}\mathcal{L}). \quad (7)$$

Compared to optimizing $\Delta$ in the original parameter space, our approach explicitly constrains the gradient directions the optimizer can take. By taking gradient steps in the direction orthogonal to the shared subspace, we narrow the gap with the task-specific model. This effectively mitigates task conflicts while retaining shared knowledge.

### 4.3. Task-aware Training-free $\lambda$

The sensitivity to $\lambda$ may arise from potential conflicts or intricate relationships among tasks, making the merging process highly dependent on the choice of this coefficient. To address this, we propose a direct method for computing task-aware $\lambda_i^l$ based solely on task vectors, thereby eliminating the need for training or additional data. Building on rethinking of the role of $\lambda$, we derive the following layer-wise, adaptive $\lambda_i^l$ calculation:

$$\lambda_i^l = \frac{\eta}{||\boldsymbol{\tau}_i^l||}, \quad \forall l \leq L, \quad (8)$$

*Table 1.* Multi-task performance when merging ViT-B/32 models on 8-task vision benchmark.

| Method | SUN397 | Cars | RESISC45 | EuroSAT | SVHN | GTSRB | MNIST | DTD | Avg. |
|---|---|---|---|---|---|---|---|---|---|
| *Non-Merging Methods* | | | | | | | | | |
| Pre-trained | 62.3 | 59.7 | 60.7 | 45.5 | 31.4 | 32.6 | 48.5 | 43.8 | 48.0 |
| Individual | 79.2 | 77.7 | 96.1 | 99.7 | 97.5 | 98.7 | 99.7 | 79.4 | 90.8 |
| Traditional MTL | 73.9 | 74.4 | 93.9 | 98.2 | 95.8 | 98.9 | 99.5 | 77.9 | 88.9 |
| *Data-Free Methods* | | | | | | | | | |
| Task Arithmetic | 55.2 | 54.9 | 66.7 | 78.9 | 80.2 | 69.7 | 97.3 | 50.4 | 69.1 |
| Ties-Merging | 59.8 | 58.6 | 70.7 | 79.7 | 86.2 | 72.1 | 98.3 | 54.2 | 72.4 |
| Consensus Merging | 65.7 | 63.6 | 76.5 | 77.2 | 81.7 | 70.3 | 97.0 | 57.1 | 73.6 |
| AWD TA | 63.5 | 61.9 | 72.6 | 84.9 | 85.1 | 79.1 | 98.1 | 56.7 | 75.2 |
| PCB-Merging | 66.7 | 65.5 | 78.5 | 79.3 | 86.4 | 77.1 | 98.2 | 59.1 | 76.3 |
| Concrete TA | 62.5 | 61.1 | 76.0 | **95.7** | **91.0** | 81.9 | **98.5** | 51.9 | 77.3 |
| **DOGE TA** (Ours) | **67.7** | **70.1** | **82.0** | 90.3 | 86.3 | **86.8** | 98.3 | **64.0** | **80.7** |
| *Test-Time Adaption Methods* | | | | | | | | | |
| AdaMerging | 64.5 | 68.1 | 79.2 | 93.8 | 87.0 | 91.9 | 97.5 | 59.1 | 80.1 |
| AdaMerging++ | 66.6 | 68.3 | 82.2 | 94.2 | 89.6 | 89.0 | 98.3 | 60.6 | 81.1 |
| Representation Surgery | 63.8 | 59.9 | 83.3 | **97.9** | 87.0 | 87.0 | 98.6 | 69.4 | 80.9 |
| AWD AM | 68.1 | 71.4 | 83.4 | 94.8 | 87.7 | 93.6 | 97.9 | 66.1 | 82.9 |
| Concrete AM | 67.8 | 70.0 | 87.5 | 96.0 | **91.6** | **96.7** | 98.7 | 63.8 | 84.0 |
| **DOGE AM** (Ours) | **70.5** | **74.8** | **88.7** | 94.1 | **91.6** | 95.7 | **98.8** | **72.5** | **85.9** |

where $L$ represents the number of layers, and $\eta$ is a hyper-parameter that sets the global magnitude. The computed $\lambda_i^l$ takes into account the differences between tasks, balancing the scale of the task vectors. By focusing on a single $\eta$, we can replace the traditional task-wise and layer-wise $\lambda$ search, reducing the risk of dominance by any single task.

To conclude, we concisely outline the pipeline of the proposed framework in Alg. 1.

## 5. Experiments

In this section, we first describe our experimental setup. Then, we present our main results. We also provide ablation studies and discussions for a thorough analysis.

### 5.1. Experimental Setup

**Datasets and pre-trained models.** For vision tasks, we use the ViT-B/32 and ViT-L/14 models, originally derived from CLIP (Radford et al., 2021). The downstream tasks encompass a variety of challenges, including SUN397 (Xiao et al., 2016), Stanford Cars (Krause et al., 2013), RE-SISC45 (Cheng et al., 2017), EuroSAT (Helber et al., 2019), SVHN (Netzer et al., 2011), GTSRB (Stallkamp et al., 2011), MNIST (LeCun, 1998), and DTD (Cimpoi et al., 2014). For NLP tasks, we use the Flan-T5-base and Flan-T5-large models (Chung et al., 2024), evaluated on eight tasks from the GLUE benchmark (Wang et al., 2019). Further details are provided in App. A.

**Implementation details.** We perform 400 iterations of learning $\Delta$ with a learning rate of $1e - 4$. The global magnitude of the merging coefficient $\eta$ is set to 0.07 for vision tasks and 0.15 for NLP tasks. The subspace basis size $k$ is

simply defined as the rank of each task vector divided by the number of tasks (*i.e.*, 8). Following Ties-Merging (Yadav et al., 2023), we retain only the top 30% of parameters with the largest magnitudes. We report Spearman's $\rho$ for STSB and the standard average accuracy (%) for other tasks. Additional information on the experimental setup for model merging can be found in App. B.

**Compared baselines.** We categorize the baselines into three main groups: Non-Merging methods, Data-Free methods, and Test-Time Adaptation methods. The non-merging category includes individually fine-tuned models and a traditional multi-task learning approach. The traditional MTL trains the base model on all tasks simultaneously, serving as an upper bound for multi-task model merging. The data-free methods we evaluate include Task Arithmetic (Ilharco et al., 2023), Ties-Merging (Yadav et al., 2023), Consensus Merging (Wang et al., 2024b), AWD TA (Xiong et al., 2024), PCB-Merging (Du et al., 2024), and Concrete TA (Tang et al., 2023). Lastly, we include TTA methods such as AdaMerging (Yang et al., 2024c) (layer-wise) and Representation Surgery (Yang et al., 2024b). Further details about these baseline methods are provided in App. C.

### 5.2. Main Results

**Vision tasks.** Tabs. 1 and 2 present the results for the ViT-B/32 and ViT-L/14 architectures, respectively. Methods like Concrete Merging and Ties-Merging address parameter conflicts by eliminating certain neurons during model merging, outperforming baselines such as TA. AdaMerging and AdaMerging++ automatically learn layer-wise merging coefficients on the test set in an unsupervised manner, also demonstrating strong performance. However, despite these

*Table 2.* Multi-task performance when merging ViT-L/14 models on 8-task vision benchmark.

| Method | SUN397 | Cars | RESISC45 | EuroSAT | SVHN | GTSRB | MNIST | DTD | Avg. |
|---|---|---|---|---|---|---|---|---|---|
| | | | | *Non-Merging Methods* | | | | | |
| Pre-trained | 66.8 | 77.7 | 71.0 | 59.9 | 58.4 | 50.5 | 76.3 | 55.3 | 64.5 |
| Individual | 82.3 | 92.4 | 97.4 | 100 | 98.1 | 99.2 | 99.7 | 84.1 | 94.2 |
| Traditional MTL | 80.8 | 90.6 | 96.3 | 96.3 | 97.6 | 99.1 | 99.6 | 84.4 | 93.5 |
| | | | | *Data-Free Methods* | | | | | |
| Task Arithmetic | 73.9 | 82.1 | 86.6 | 94.1 | 87.9 | 86.7 | 98.9 | 65.6 | 84.5 |
| Ties-Merging | 76.5 | 85.0 | 89.3 | 95.7 | 90.3 | 83.3 | 99.0 | 68.8 | 86.0 |
| Consensus Merging | 75.0 | 84.3 | 89.4 | 95.6 | 88.3 | 82.4 | 98.9 | 68.0 | 85.2 |
| AWD TA | 76.2 | 85.4 | 88.7 | 96.1 | 92.4 | 92.3 | **99.3** | 69.4 | 87.5 |
| PCB-Merging | 76.8 | 86.2 | 89.4 | **96.5** | 88.3 | 91.0 | 98.6 | 73.6 | 87.5 |
| Concrete TA | **86.2** | 66.9 | **96.7** | 93.4 | **99.1** | 89.0 | 74.6 | **93.6** | 87.4 |
| DOGE TA (Ours) | 76.7 | **87.7** | 91.6 | 96.2 | 94.4 | **93.4** | 98.9 | 71.6 | **88.8** |
| | | | | *Test-Time Adaption Methods* | | | | | |
| AdaMerging | 79.0 | 90.3 | 90.8 | 96.2 | 93.4 | 98.0 | 99.0 | 79.9 | 90.8 |
| AdaMerging++ | 79.4 | 90.3 | 91.6 | 97.4 | 93.4 | 97.5 | 99.0 | 79.2 | 91.0 |
| Representation Surgery | 75.7 | 84.4 | 93.1 | **98.8** | 91.3 | 93.4 | 99.1 | 76.1 | 89.0 |
| AWD AM | **79.8** | 90.6 | 91.8 | 97.0 | 93.9 | 98.4 | **99.2** | 81.1 | 91.5 |
| Concrete AM | 77.8 | 91.2 | 92.1 | 97.0 | 94.4 | 97.9 | 99.0 | 79.5 | 91.1 |
| DOGE AM (Ours) | 79.7 | **91.6** | **94.4** | 96.7 | **96.5** | **98.6** | 99.0 | **84.1** | **92.6** |

*Table 3.* Multi-task performance when merging Flan-T5-base (LoRA fine-tuned) models on all eight tasks.

| Method | CoLA | MNLI | MRPC | QNLI | QQP | RTE | SST2 | STSB | Avg. |
|---|---|---|---|---|---|---|---|---|---|
| Individual | 69.1 | 82.7 | 85.5 | 90.9 | 84.0 | 84.4 | 92.9 | 87.4 | 84.6 |
| | | | | *Data-Free Methods* | | | | | |
| Weight Averaging | **69.7** | 59.7 | 78.9 | 90.1 | **83.8** | **80.5** | 91.2 | 72.0 | 78.2 |
| Task Arithmetic | 68.8 | 55.2 | 78.7 | 89.8 | 83.7 | 79.1 | 91.5 | 72.4 | 77.4 |
| Ties-Merging | 68.3 | 56.3 | 79.4 | 89.8 | 83.7 | 79.4 | 91.6 | 71.2 | 77.5 |
| Concrete TA | 69.1 | 58.1 | 78.4 | 89.9 | 83.5 | 79.4 | 91.6 | 73.4 | 78.0 |
| DOGE TA (Ours) | 69.1 | **71.9** | **80.9** | **90.3** | 83.5 | 79.8 | **92.5** | 71.1 | **79.9** |
| | | | | *Test-Time Adaption Methods* | | | | | |
| AdaMerging++ | 69.1 | 60.3 | 78.4 | 90.0 | 83.6 | 79.1 | 91.6 | 74.1 | 78.3 |
| Concrete AM | 69.0 | 59.4 | 80.1 | 89.9 | 82.9 | 79.1 | 91.7 | **75.4** | 78.5 |

advances, all existing model merging methods still show a noticeable gap compared to individually fine-tuned models. AWD also optimizes Δ but focuses on increasing orthogonality among task vectors, neglecting the performance gap with individually fine-tuned models. In contrast, our proposed DOGE is orthogonal to existing merging methods and can complement them. When applied to Task Arithmetic and AdaMerging, significant performance improvements are observed. For instance, on ViT-B/32, Task Arithmetic's accuracy improves from 69.1% to 80.7% with DOGE. For the test-time adaptation method AdaMerging, accuracy increases from 80.1% to 85.9%. On ViT-L/14, AdaMerging achieves 92.6% accuracy after incorporating DOGE, nearly matching the 93.5% achieved by Traditional MTL.

**Language tasks.** We extend our approach to language models and LoRA fine-tuned models to evaluate its generalizability (Li et al., 2023). Unlike classification tasks, text-to-text generation requires generating coherent outputs rather than merely projecting hidden representations to logits, introducing additional complexity (Li et al., 2025b). Tabs. 3 and 4 present the results on Flan-T5-base and Flan-

T5-large models. Given that pre-trained LLMs already exhibit strong multitasking capabilities, the potential for substantial improvement via specialized methods is inherently limited. Nevertheless, our approach achieves the highest performance, even outperforming TTA methods under data-free conditions. On Flan-T5-large, our data-free method achieves an accuracy of 88.0%, closely approaching the performance of individually fine-tuned models at 89.6%. These results highlight the superior generalization ability of our method across diverse models and tasks.

### 5.3. Ablation Studies

**Generalization and robustness evaluation.** To further assess the generalization and robustness of our approach, we conduct experiments on unseen tasks and corrupted test sets (*i.e.*, out-of-distribution). Tab. 5 presents generalization results on two unseen tasks. On in-domain tasks, our approach (under data-free conditions) performs comparably to AdaMerging, which leverages the test set for adaptation. Notably, on unseen tasks, where no corresponding task vectors were merged, our method outperforms AdaMerging by

Table 4. Multi-task performance when merging Flan-T5-large (LoRA fine-tuned) models on all eight tasks.

| Method | CoLA | MNLI | MRPC | QNLI | QQP | RTE | SST2 | STSB | Avg. |
|---|---|---|---|---|---|---|---|---|---|
| Individual | 80.2 | 88.5 | 89.2 | 94.4 | 87.2 | 91.7 | 95.2 | 90.9 | 89.6 |
| *Data-Free Methods* | | | | | | | | | |
| Weight Averaging | 74.6 | 84.3 | 84.1 | 92.8 | 86.3 | 87.4 | 94.8 | 88.0 | 86.5 |
| Task Arithmetic | 76.9 | 85.4 | 85.3 | **93.9** | 85.8 | 88.1 | 95.2 | 87.8 | 87.3 |
| Ties-Merging | 77.1 | 85.1 | 86.3 | **93.9** | 86.0 | 87.7 | 95.1 | 88.0 | 87.4 |
| Concrete TA | 76.6 | 86.4 | 86.0 | **93.9** | 85.9 | **88.4** | 95.2 | 87.9 | 87.5 |
| DOGE TA (Ours) | **78.4** | **88.1** | **86.5** | 93.8 | **86.3** | 87.7 | 95.1 | 87.7 | **88.0** |
| *Test-Time Adaption Methods* | | | | | | | | | |
| AdaMerging++ | 76.7 | 87.6 | 84.8 | 93.8 | 85.9 | 88.1 | 95.2 | **88.6** | 87.6 |
| Concrete AM | 76.1 | 87.7 | 85.5 | 93.8 | 85.9 | 88.1 | **95.4** | 87.1 | 87.5 |

Table 5. Generalization results on two unseen tasks when merging ViT-B/32 models on six tasks.

| Method | Seen Tasks | | | | | | | Unseen Tasks | | |
|---|---|---|---|---|---|---|---|---|---|---|
| | SUN397 | Cars | RESISC45 | DTD | SVHN | GTSRB | Avg. | MNIST | EuroSAT | Avg. |
| Pre-trained | 63.2 | 59.9 | 60.6 | 43.9 | 23.5 | 30.4 | 46.9 | 47.6 | 45.6 | 46.6 |
| Task Arithmetic | 64.3 | 63.0 | 73.2 | 54.9 | 84.7 | 79.5 | 69.9 | 75.5 | 42.6 | 59.1 |
| Ties-Merging | 68.3 | 65.5 | 76.9 | 54.9 | 75.4 | 72.0 | 68.9 | 73.1 | 47.3 | 60.2 |
| AdaMerging | 68.4 | 71.9 | **87.9** | **69.1** | **92.2** | **93.8** | **80.5** | 77.7 | 47.3 | 62.5 |
| DOGE TA (Ours) | **69.8** | **72.6** | 86.6 | 67.6 | 90.8 | 91.6 | 79.8 | **81.3** | **48.2** | **64.8** |

an average of 2.3%, demonstrating superior generalization. By contrast, TTA methods rely on the test set, which constrains their ability to generalize. Furthermore, Tab. 11 in Appendix evaluates each method's robustness on corrupted test sets, designed to simulate real-world scenarios where input data may be noisy or corrupted. The results underline our approach's overall strength and efficacy, particularly in handling noise and out-of-distribution data.

**Effects of each module.** Tab. 6 evaluates the contribution of each module to overall performance. We start with DOGE TA and remove one component at a time, reporting the performance for full model merging (ViT-B/32) and for merging PEFT models (T5-base on GLUE). Removing $\Delta$ optimization corresponds to using the task-aware $\lambda$ on TA, underscoring the effectiveness of the data-free objective applied to task vectors, which reduces conflicts between tasks. In cases where the shared subspace is removed, $\Delta$ optimization occurs in the original parameter space. This demonstrates that optimizing within the shared subspace enables the merged model to capture shared knowledge across multiple tasks. When task-aware $\lambda$ is removed, we utilize a uniform merging coefficient of 0.3. Tab. 13 in Appendix further presents the task-wise and layer-wise improvements over TA. Tab. 6 shows that each component is crucial for achieving optimal performance; particularly, $\Delta$ optimization and the shared subspace are

Table 6. Effects of the proposed modules.

| Model | ViT-B/32 | T5-base |
|---|---|---|
| Task Arithmetic | 69.1 | 77.4 |
| DOGE TA | **80.7** | **79.9** |
| $-\Delta$ Optimization | 71.9 | 77.9 |
| $-$ Shared Subspace | 77.2 | 79.0 |
| $-$ Task-Aware $\lambda$ | 79.2 | 79.8 |

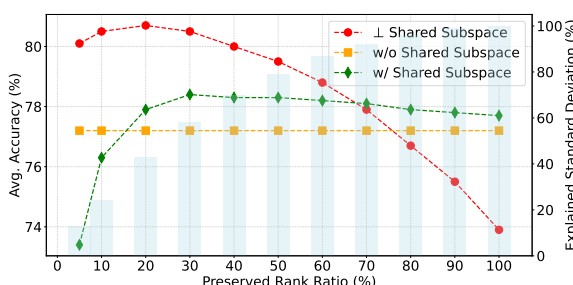

Figure 4. The average accuracy changes corresponding to different rank ratios in the subspace under ViT-B/32 architecture.

most vital, causing notable performance drops of 8.8% and 3.5% in vision tasks, and 2.0% and 0.9% in language tasks, respectively. With all modules included, we achieve the best performance, boosting TA by 5%-11% and demonstrating the complementarity of these components.

**Effects of the subspace.** Since the effectiveness of our method hinges on the decomposition of the subspace, we explore the impact of the rank ($k$ of $S_{\text{share}}$) on merging performance. Fig. 4 displays the performance with varying rank ratios, alongside the explained standard deviation (*i.e.*, the ratio of preserved singular values $\sigma$ to the total sum of singular values $\Sigma$). Updates performed orthogonally to the subspace direction have shown positive results, with the optimal rank identified between 10%-30%, where the explained standard deviation already exceeds 40%. Preserving a higher rank introduces noise, resulting in a high volume of constraints in the gradient space. Updates along the direction of the shared subspace also slightly outperform those in the original parameter space, due to the allowance for

*Table 7.* Different gradient projection directions in the subspace when merging ViT-B/32 models.

| Method | SUN397 | Cars | RESISC45 | EuroSAT | SVHN | GTSRB | MNIST | DTD | Avg Acc |
|---|---|---|---|---|---|---|---|---|---|
| Pre-trained | 62.3 | 59.7 | 60.7 | 45.5 | 31.4 | 32.6 | 48.5 | 43.8 | 48.0 |
| Individual | 79.2 | 77.7 | 96.1 | 99.7 | 97.5 | 98.7 | 99.7 | 79.4 | 90.8 |
| ⊥ Shared Subspace | **67.7** | **70.1** | **82.0** | **90.3** | 86.3 | **86.8** | **98.3** | **64.0** | **80.7** |
| w/o Shared Subspace | 63.3 | 67.1 | 74.9 | 85.2 | 86.9 | 83.9 | 98.2 | 57.9 | 77.2 |
| w/ Shared Subspace | 62.2 | 66.6 | 74.7 | 87.3 | **88.7** | 84.7 | **98.3** | 57.5 | 77.5 |

*Table 8.* Sensitivity analysis for the global scaling factor $\eta$.

| $\eta$ | 0.01 | 0.02 | 0.03 | 0.04 | 0.05 | 0.06 | 0.07 | 0.08 | 0.09 |
|---|---|---|---|---|---|---|---|---|---|
| ViT-B/32 | 79.5 | 80.3 | 80.6 | 80.9 | **81.0** | 80.8 | 80.7 | 80.2 | 79.8 |

*Table 9.* The computational time and GPU memory requirements for optimizing $\Delta$ in the subspace.

| Model | Solving Time | GPU Memory |
|---|---|---|
| ViT-B/32 | 121s | 729MB |
| ViT-L/14 | 311s | 2448MB |

*Table 10.* Normalized scores are computed relative to individual models when merging WizardLM-13B (Instruction-Following), WizardMath-13B (Math), and LLaMA-2-13B-code-alpaca (Code).

| Method | AlpacaEval | GSM8K | MATH | HumanEval | MBPP | Avg. |
|---|---|---|---|---|---|---|
| Individual | 100.0 | 100.0 | 100.0 | 100.0 | 100.0 | 100.0 |
| TA | 102.7 | 91.0 | 70.5 | 50.0 | 87.7 | 80.4 |
| TIES | 98.1 | 97.4 | 68.1 | 60.0 | 89.4 | 82.6 |
| TA + DARE | 103.1 | 88.0 | 72.5 | 63.3 | 92.9 | 84.0 |
| TIES + DARE | **107.9** | 90.3 | 65.6 | **80.0** | 92.4 | 87.2 |
| **Ours** | 107.5 | **105.0** | **94.4** | 56.7 | 86.5 | **90.0** |

learning personalized subspaces. Tab. 7 reports the specific performance across eight tasks at the same rank. Compared to w/o or updates only along the subspace direction, we observe significant improvements on the DTD dataset but decreased performance on the SVHN dataset. This is attributable to DTD's requirement for rich textural and geometric features, which are well-preserved in the shared subspace. Conversely, SVHN (Street View House Numbers) differs significantly in visual representation from other tasks, making the primary components in the shared subspace less suitable for SVHN. This is further evidenced by the gap from the pre-trained model to individual performance: SVHN shows the lowest pre-trained performance at 31.4%, yet finetuning results peak at 97.5%, indicating a need for task-specific features. In summary, this demonstrates that our method effectively preserves shared knowledge across multiple tasks, achieving optimal overall performance.

**Hyperparameter sensitivity.** Additional sensitivity analysis for the global scaling factor $\eta$ is provided in Tab. 8. Evaluations across $\eta$ values from 0.01 to 0.09 show that performance remains stable, even reaching higher values at certain points. (We did not conduct an exhaustive grid search; this range was chosen because the computed $\eta$ was close to 0.03.) This consistent performance across different $\eta$ values demonstrates the robustness of our approach and highlights the practicality of task-aware coefficients.

**Computational requirements.** As illustrated in Tab. 9, our approach involves optimizing $\Delta$ within the subspace across 8 vision tasks over 400 iterations. The results demonstrate that our method incurs minimal computational overhead across different model variants and requires only moderate GPU memory. This efficiency is achieved through layer-wise optimization and fast convergence via gradient descent. Notably, the SVD operation is performed only

once at the beginning, with a computational complexity of $O(\min(mn^2, m^2n))$. These findings highlight the near-universal scalability of our method on devices equipped with modern GPUs.

**Generative language tasks.** We further extend our method to LLMs and conduct experiments following standard settings (Yu et al., 2024). The merging process is completed in just 58 minutes on a single A100 GPU. We report normalized scores relative to the performance of individual models when merging WizardLM-13B (Instruction-Following), WizardMath-13B (Math), and llama-2-13b-code-alpaca (Code). As shown in Tab. 10, our method achieves the highest average performance across tasks, demonstrating its effectiveness and scalability in generative language tasks.

## 6. Conclusion

Existing merging methods often prioritize mitigating task conflicts, neglecting a critical requirement of model merging: achieving performance comparable to task-specific models. In this paper, we rethink model merging from a multi-task learning perspective, treating it as a constrained optimization problem. We introduce an adaptive projective gradient descent method that optimizes a data-free objective within a shared subspace and includes adaptive merging coefficients. Extensive experiments validate the superior generalization and robustness of our approach, highlighting its effectiveness across various benchmarks.

## Acknowledgments

This work is supported by the National Key R&D Program of China (2022YFB4701400/4701402), SSTIC Grant (KJZD20230923115106012, KJZD20230923114916032, GJHZ20240218113604008), Beijing Key Lab of Networked

Multimedia, the Shenzhen Basic Research Project (Natural Science Foundation) Basic Research Key Project (NO. JCYJ20241202124430041), National Natural Science Foundation of China (No. 62025604).

## Impact Statement

The use of large-scale image datasets often involves privacy, labor, and ethical challenges, limiting research opportunities. As a result, the research community is turning towards leveraging pre-trained models. Model merging offers a novel approach to multi-task learning by utilizing the abundant expert models made available by the open-source ethos. With more than a million models accessible on Hugging Face, this strategy leverages the community's vast resources. This shift enables the creation of multi-task models by directly merging independently trained expert models without needing original training data, presenting a new paradigm.

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

# A. Model Details

For vision tasks, we employ pre-trained models from CLIP (Radford et al., 2021), fine-tuning them using the AdamW optimizer with a weight decay of 0.1 and a learning rate of $1 \times 10^{-5}$. The downstream tasks encompass a variety of challenges. SUN397 (Xiao et al., 2016) is a large-scale scene recognition dataset comprising over 100,000 images across 397 indoor and outdoor scene categories. Stanford Cars (Krause et al., 2013) contains 16,185 images of 196 car models and is commonly used for fine-grained image classification. RESISC45 (Cheng et al., 2017) consists of 31,500 remote sensing images evenly distributed over 45 scene categories, supporting research in aerial scene classification. EuroSAT (Helber et al., 2019) is based on Sentinel-2 satellite images and includes 27,000 samples covering 10 land use and land cover classes. SVHN (Netzer et al., 2011) is a real-world digit recognition dataset with over 600,000 images of house numbers captured from Google Street View. GTSRB (Stallkamp et al., 2011) comprises more than 50,000 images of 43 traffic sign categories, serving as a benchmark for traffic sign recognition tasks. MNIST (LeCun, 1998) is a well-known dataset for handwritten digit classification, featuring 70,000 grayscale images of digits from 0 to 9. DTD (Cimpoi et al., 2014) is a texture dataset with 5,640 images organized into 47 human-describable categories, designed for studying texture perception and classification. We measure the models' performance using top-1 accuracy as the primary metric (Horoi et al., 2024; Stoica et al., 2024; Wei et al., 2025; Guan et al., 2024).

For NLP tasks, our pre-trained model is Flan-T5 (Wang et al., 2024a). We deploy Flan-T5 on eight tasks from the GLUE benchmark (Wang et al., 2019), including CoLA, MNLI, MRPC, QNLI, QQP, RTE, SST2, and STSB. To ensure consistency and reproducibility, we use the same parameter-efficient models following FusionBench (Tang et al., 2024a). The Flan-T5 models, which are encoder-decoder Transformer models, undergo LoRA fine-tuning with hyperparameters $r = 16$ and $\alpha = 32$ (Hu et al., 2022). We maintain a constant learning rate of $4 \times 10^{-5}$ and a uniform batch size of 16 across all tasks, fine-tuning for 2000 steps per task. Adapting to the text-to-text framework, we have restructured the initial inputs accordingly. Performance is evaluated using exact match accuracy for all tasks, except for STSB where we report Spearman's $\rho$.

# B. Implementation Details

The experiments in our study were conducted on a consistent hardware setup, utilizing NVIDIA GTX 4090 GPUs equipped with 24GB of memory. We performed 400 iterations of learning $\Delta$ with a learning rate of $1e - 4$ using the Adam optimizer. The global magnitude of the merging coefficient $\eta$ is set to 0.07 for vision tasks and 0.15 for NLP tasks. We did not perform a specialized grid search. This setting was chosen because the calculated average $\lambda$ was close to 0.3, which is a beneficial scaling coefficient for the Task Arithmetic method, demonstrating that our approach is not tricky. The subspace basis size $k$ is simply defined as the rank of the task vector divided by the number of tasks (*i.e.*, 8), with the shared subspace basis size set at the rank divided by 6. Following Ties-Merging (Yadav et al., 2023), we retain only the top 30% of parameters with the largest magnitudes. We only apply our method to the linear layer in the model. For the implementation of our experiments, we employed PyTorch version 2.5 with Python 3.10.

# C. Compared Baselines

**Pre-trained**: Uses a pre-trained model for each task without integrating task-specific information. Serves as a basic benchmark for comparison.

**Individual**: Fine-tunes a separate model for each task, ensuring no task interference and providing an ideal baseline for task-specific performance.

**Traditional MTL**: Trains a single base model on all tasks simultaneously, representing the upper bound for multi-task learning.

**Weight Averaging** (Wortsman et al., 2022): Simply averages the weights of models fine-tuned on different tasks without considering task-specific dynamics.

**Task Arithmetic** (Ilharco et al., 2023): Computes task vectors for individual tasks and sums them up to construct a multi-task vector. This vector is scaled by a coefficient ($\lambda$) and added to the pre-trained model's initial parameters.

**Fisher Merging** (Matena & Raffel, 2022): Uses the Fisher information matrix to assess parameter importance, guiding the merging process to retain critical parameters for each task.

**Ties-Merging** (Yadav et al., 2023): Combines steps like trimming, parameter sign determination, and disjoint merging to

produce a merged task vector $\tau$. The final model is defined as $\theta = \theta_0 + \lambda\tau$, where $\lambda$ is tuned using a validation set.

**Consensus Merging** (Wang et al., 2024b): Improves traditional merging methods by removing "selfish" and "catastrophic" weights—parameters beneficial only to specific tasks but detrimental to others.

**AWD (Adaptive Weight Disentanglement)** (Xiong et al., 2024): Enhances orthogonality among task vectors to minimize interference and improve multi-task merging.

**PCB-Merging** (Du et al., 2024): Combines intra-balancing, which evaluates the significance of parameters within individual tasks, and inter-balancing, which measures parameter similarities across tasks. Parameters with low importance scores are pruned, while the remaining parameters are rescaled to create the final merged model.

**Concrete Merging** (Tang et al., 2023): Introduces a meta-learning framework to generate a concrete mask for mitigating task interference.

**AdaMerging** (Yang et al., 2024c): Learns task-wise or layer-wise merging coefficients adaptively using entropy minimization on unlabeled test data as a surrogate objective.

**AdaMerging++** (Yang et al., 2024c): Extends AdaMerging by incorporating task vector adjustments from Ties-Merging, removing parameter redundancies, and resolving sign conflicts.

**Representation Surgery** (Yang et al., 2024b): Aligns the representation of the merged model with independent models while calibrating biases to ensure task compatibility.

# D. Experiment Results

Table 11. Robustness to the test data distribution on ViT-B/32.

| Method | Cars | EuroSAT | RESISC45 | GTSRB | Avg. | Cars | EuroSAT | RESISC45 | GTSRB | Avg. |
|---|---|---|---|---|---|---|---|---|---|---|
| | | | Clean Test Set | | | | | Corrupted Test Set (Motion Blur) | | |
| Fisher Merging | 66.0 | 92.7 | 83.7 | 78.7 | 80.3 | 60.7 | 57.6 | 81.7 | 78.4 | 69.6 |
| Task Arithmetic | 64.6 | 91.8 | 80.2 | 74.8 | 77.9 | 62.4 | 59.2 | 78.5 | 63.3 | 65.9 |
| Ties-Merging | 65.2 | 83.3 | 78.1 | 67.4 | 73.5 | 64.4 | 53.9 | 76.4 | 57.1 | 62.9 |
| AdaMerging | 75.2 | 94.3 | 87.6 | 96.7 | 88.5 | 72.4 | 72.7 | 85.3 | 94.3 | 81.2 |
| DOGE TA | 72.5 | 95.6 | 86.4 | 90.3 | 86.2 | 70.7 | 71.7 | 85.0 | 82.7 | 77.5 |
| DOGE AM | **77.3** | **96.4** | **91.5** | **97.6** | **90.7** | **74.7** | **79.1** | **89.5** | **95.7** | **84.8** |
| | | | Corrupted Test Set (Impluse Noise) | | | | | Corrupted Test Set (Gaussian Noise) | | |
| Fisher Merging | 61.5 | 50.0 | 74.7 | 52.6 | 59.7 | 61.6 | 48.1 | 76.0 | 51.3 | 59.3 |
| Task Arithmetic | 59.8 | 53.3 | 72.3 | 45.0 | 57.6 | 61.5 | 52.5 | 75.0 | 50.1 | 59.8 |
| Ties-Merging | 60.2 | 45.6 | 69.8 | 38.3 | 53.5 | 61.8 | 47.3 | 73.1 | 42.3 | 56.1 |
| AdaMerging | **69.2** | 40.0 | **79.6** | 83.3 | **68.0** | 70.0 | **53.3** | 82.1 | 80.0 | 71.4 |
| DOGE TA | 66.7 | **57.2** | 79.2 | 61.0 | 66.0 | 68.7 | 50.9 | 81.7 | 64.1 | 66.4 |
| DOGE AM | 68.6 | 25.7 | **79.6** | **86.5** | 65.1 | **71.2** | 50.7 | **86.2** | **83.2** | **72.8** |
| | | | Corrupted Test Set (Pixelate) | | | | | Corrupted Test Set (Spatter) | | |
| Fisher Merging | 2.2 | 34.0 | 17.0 | 63.2 | 29.1 | 61.4 | **64.2** | 74.6 | 47.3 | 61.9 |
| Task Arithmetic | 2.3 | 33.2 | 19.1 | 65.6 | 30.0 | 61.0 | 62.5 | 72.8 | 57.0 | 63.3 |
| Ties-Merging | 3.3 | 31.8 | 18.0 | 58.5 | 27.9 | 61.3 | 52.9 | 70.3 | 48.1 | 58.2 |
| AdaMerging | 1.3 | 52.9 | 21.0 | 91.0 | 41.5 | 68.4 | 55.9 | 78.3 | 92.3 | 73.7 |
| DOGE TA | **3.4** | 39.9 | 21.9 | 84.6 | 37.5 | 68.4 | 63.9 | 78.9 | 75.3 | 71.6 |
| DOGE AM | 1.4 | **55.9** | **25.8** | **93.4** | **44.1** | **71.0** | 54.5 | **83.9** | **93.4** | **75.7** |
| | | | Corrupted Test Set (Contrast) | | | | | Corrupted Test Set (JPEG Compression) | | |
| Fisher Merging | 63.8 | 58.4 | 75.5 | 70.4 | 67.0 | 66.3 | 67.6 | 82.6 | 58.9 | 68.8 |
| Task Arithmetic | 62.3 | 55.7 | 75.3 | 70.8 | 66.0 | 63.9 | 66.1 | 80.1 | 61.0 | 67.8 |
| Ties-Merging | 64.2 | 52.4 | 74.8 | 63.5 | 63.7 | 65.0 | 59.5 | 77.9 | 53.2 | 63.9 |
| AdaMerging | 73.1 | 67.4 | 83.0 | 96.2 | 79.9 | 72.9 | 70.7 | 86.3 | 90.6 | 80.1 |
| DOGE TA | 70.2 | 66.3 | 82.1 | 86.8 | 76.4 | 71.8 | 76.4 | 86.3 | 76.9 | 77.9 |
| DOGE AM | **75.1** | **73.5** | **87.9** | **96.9** | **83.4** | **75.0** | **78.1** | **90.0** | **92.4** | **83.9** |

**Robustness.**    To evaluate our approach's robustness to real-world variations, where data characteristics can significantly differ, we conducted extensive ablation studies across diverse data distributions. These studies specifically assessed the model's performance on out-of-distribution (OOD) data (Zhang et al., 2024a;b; Dong et al., 2023a;b; Zhang et al., 2025). To simulate real-world conditions, we introduced various types of noise into the test data following the procedure outlined by Yang et al. (2024c). Eight distinct noise types were used—motion blur, impulse noise, Gaussian noise, pixelation, spatter, contrast, and JPEG compression—to reflect a wide range of potential distortions encountered in practical applications.

The test sets included both clean and corrupted conditions to emulate distribution shifts. As shown in Tab. 11, while each strategy exhibited varying levels of robustness to different distortions, our approach consistently achieved the highest accuracy across most scenarios, often by a notable margin. Notably, DOGE AM demonstrated exceptional resilience under severe conditions such as pixelation and spatter, significantly outperforming other methods. This consistent performance across diverse corruptions underscores DOGE AM's robustness and adaptability, making it particularly effective for challenging OOD environments in real-world applications.

We conduct experiments evaluating generalization on three unseen tasks when merging five other tasks. The results in Tab. 12 reveal that SUN397, DTD, and Cars datasets pose challenges for ViT models, while MNIST/EuroSAT show limited generalization to these complex tasks (Wei et al., 2024b). Despite this, our method consistently outperformed other model merging approaches by a significant margin.

*Table 12.* Generalization results on three unseen tasks when merging ViT-B/32 models on five tasks.

| Method | Seen Tasks | | | | | | Unseen Tasks | | | |
|---|---|---|---|---|---|---|---|---|---|---|
| | RESISC45 | SVHN | GTSRB | MNIST | EuroSAT | **Avg.** | SUN397 | Cars | DTD | **Avg.** |
| Pre-trained | 60.6 | 23.5 | 30.4 | 47.6 | 45.6 | 41.5 | 63.2 | 59.9 | 43.9 | 55.6 |
| Task Arithmetic | 52.8 | 83.9 | 71.1 | 97.7 | 61.9 | 73.5 | 27.9 | 25.0 | 26.4 | 26.4 |
| Ties-Merging | 74.6 | 89.1 | 81.8 | 97.7 | 73.7 | 83.4 | 57.5 | 51.9 | 38.7 | 49.4 |
| AdaMerging | 73.5 | 76.0 | 81.5 | 97.4 | 69.4 | 79.6 | 42.3 | 37.8 | 32.0 | 37.4 |
| **DOGE TA (Ours)** | **82.6** | **89.4** | **89.0** | **98.6** | **92.3** | **90.4** | **58.7** | **54.3** | **41.4** | **51.5** |

**Effects of $\lambda$.**    Tab. 13 compares our two proposed variants of task-aware and layer-wise $\lambda$ with the baseline Task Arithmetic. We observe that applying task-wise $\lambda$ provides a noticeable improvement over the baseline, boosting the average accuracy from 69.1% to 70.7%. Further refining the granularity to layer-wise $\lambda$ achieves a new highest average accuracy of 71.9%.

*Table 13.* Task-aware and training-free $\lambda$ combined with Task Arithmetic.

| Method | SUN397 | Cars | RESISC45 | EuroSAT | SVHN | GTSRB | MNIST | DTD | Avg. |
|---|---|---|---|---|---|---|---|---|---|
| Task Arithmetic | 55.2 | 54.9 | 66.7 | 78.9 | **80.2** | **69.7** | **97.3** | 50.4 | 69.1 |
| + Task-wise $\lambda$ | 61.4 | 62.5 | 70.0 | 82.8 | 71.3 | 66.4 | 95.1 | 56.1 | 70.7 |
| + Layer-wise $\lambda$ | **62.6** | **63.9** | **71.0** | **86.8** | 73.2 | 65.2 | 95.9 | **56.4** | **71.9** |

**More task numbers.**    Tab. 14 illustrates the robustness of our approach when handling a larger number of tasks. Following Wang et al. (2024b), we evaluate its performance as more tasks are merged. In addition to the previously used 8 tasks, the 14-task scenario incorporates CIFAR100, STL10, Flowers102, OxfordIIITPet, PCAM, and FER2013. The 20-task scenario further adds six tasks: EMNIST, CIFAR10, Food101, FashionMNIST, RenderedSST2, and KMNIST. Our approach exhibits increasingly significant performance advantages as the number of tasks grows, demonstrating its effectiveness in mitigating negative transfer through gradient descent while preserving task-specific knowledge.

**Comparisons with dynamic merging.**    As shown in Tab. 15, merging multiple models into a single model presents notable challenges. DOGE is a static, plug-and-play merging method (similar to Task Arithmetic) that maintains the standard model size and supports parallelized inference. In contrast, dynamic merging approaches (Tang et al., 2024b; Huang et al., 2024; Lu et al., 2024) offer greater flexibility by dynamically selecting task-specific modules but typically require additional storage and encounter scalability considerations during inference. These methods often rely on either dynamic I/O loading of modules or maintaining all components in GPU memory. For instance, some methods train routing networks using validation data to guide module selection.

*Table 14.* Average accuracy (%) when merging models across a larger number of tasks.

| | Method | ViT-B/32 | | | ViT-L/14 | | |
|---|---|---|---|---|---|---|---|
| | | 8 tasks | 14 tasks | 20 tasks | 8 tasks | 14 tasks | 20 tasks |
| *Data-Free* | Pre-trained | 48.4 | 57.3 | 56.1 | 64.4 | 68.0 | 65.1 |
| | Weight averaging | 66.5 | 64.4 | 61.1 | 79.4 | 76.6 | 71.5 |
| | Task Arithmetic | 70.8 | 65.4 | 60.6 | 84.8 | 79.3 | 74.0 |
| | TIES | 75.1 | 68.0 | 63.4 | 86.9 | 79.5 | 75.7 |
| | Consensus TA | 75.0 | 70.4 | 65.4 | 86.2 | 82.2 | 78.9 |
| | Consensus TIES | 74.8 | 67.7 | 63.2 | 86.9 | 81.5 | 76.8 |
| | **DOGE TA** (Ours) | 80.7 (+5.7%) | 77.9 (+7.5%) | 72.5 (+7.1%) | 88.8 (+2.6%) | 87.1 (+4.9%) | 81.0 (+2.1%) |

*Table 15.* Distinction based on parameters, data requirements, and computational costs.

| Method | Parameters | Router | Data | Parallel | Performance |
|---|---|---|---|---|---|
| Task Arithmetic | $1\times$ | - | - | static | 69.1 |
| AdaMerging | $1\times$ | - | unlabeled test dataset | static | 80.1 |
| DOGE TA | $1\times$ | - | - | static | 81.0 ($\uparrow$**11.6**) |
| DOGE AM | $1\times$ | - | unlabeled test dataset | static | 85.9 ($\uparrow$**5.8**) |
| Representation Surgery | $> 1\times$ | - | unlabeled test dataset | static | 80.9 |
| EMR merging | $4\times$ | perfect router | - | dynamic | 88.7 |
| Twin merging | $2.25\times$ | trained router | labeled validation dataset | dynamic | 86.1 |
| WEMoE | $5\times$ | trained router | unlabeled test dataset | dynamic | 89.4 |
| Traditional MTL | $1\times$ | - | - | - | 88.9 |
| Multiple Models | $8\times$ | - | - | - | 90.8 |

**Potential limitations** A potential limitation is the lack of consideration for heterogeneous model merging, which requires transformation when task vectors have inconsistent shapes or layer numbers.

