# OpenReview forum: "Modeling Multi-Task Model Merging as Adaptive Projective Gradient Descent"
_ICML.cc/2025/Conference — ICML 2025 poster_

### Official Review · Reviewer_tyLA · 2025-02-22

**Overall Recommendation:** 3

**Summary:**

This paper views model merging from a multi-task learning angle. It designs an adaptive projective gradient descent method that tries to minimize the gap between the merged model and individual models, subject to the constraint of retaining shared knowledge. Specifically, the method only uses gradients in the orthogonal direction of the shared space of task vectors. Experiments show performance improvement compared to baseline methods.

## update after rebuttal
As discussed during the rebuttal, I generally support the acceptance of this paper despite its performance gap issue. The proposal should be helpful for some other researchers working in this field.

**Claims And Evidence:**

The main goal of the paper, which is ambitious, is "ensuring the merged model performs comparably to task-specific models on respective tasks". This is not well supported by the experimental results. Based on Tables 1, 2, and 3, the model merging proposal does not maintain the same level of performance as the individually trained ones, and is even worse than multi-task learning in many cases.

Theoretically, it is also unclear why "only take gradient steps in the direction orthogonal to the shared space" (Line 68) can help us achieve this ambitious goal. The argument is not convincing.

**Essential References Not Discussed:**

I am not aware of closely related works that were not discussed in the paper.

**Experimental Designs Or Analyses:**

The experimental designs are satisfactory, comparing the proposal with SOTA methods and analyzing different modules of the method.

**Methods And Evaluation Criteria:**

The proposed method needs more justification to help me understand why it can help keep task-specific information while model merging.

The experimental evaluations are diverse and the proposal shows better performance compared to previous model merging methods.

**Other Comments Or Suggestions:**

NA

**Other Strengths And Weaknesses:**

**Other strengths:**

- This paper is interesting and shows promising performance compared to previous methods based on the experimental results.
- The manuscript is well-written, with a proper discussion of previous works.

**Other weakness:**

- The connection between the motivation (keeping task-specific information) and the method is not very clear.

**Questions For Authors:**

NA

**Relation To Broader Scientific Literature:**

This paper is related to many works on model merging and multi-task learning.

**Theoretical Claims:**

The paper does not provide much theoretical evidence.

---

> ### Author Rebuttal · Authors · 2025-03-31
>
> Thanks for your review and detailed comments. We hope the following discussion can address your concerns!
> ___
> > Q1: Based on Tables 1, 2, and 3, the model merging proposal does not maintain the same level of performance as the individually trained ones, and is even worse than multi-task learning in many cases.
>
> A1: Transfer learning has driven the proliferation of fine-tuned models, **but deploying separate models for each task creates significant storage and computational burdens.** While multi-task learning could address this, it involves costly training and simultaneous access to all tasks. Additionally, determining the optimal data mixture for effective multi-task training can be complex and resource-intensive. Model merging addresses these challenges by **compressing** task-specific models without requiring access to training data (privacy or copyright). The performance gap between merged model and individual models or multi-task learning is an inherent constraint, as merging multiple trained models into a single model occurs without the benefit of costly computations.
>
> While previous methods focus on alleviating conflicts between tasks, our approach takes a more direct path by establishing the minimization of the gap between the merged model and individual models as our explicit optimization objective (Line 24). By formulating and effectively solving this as a data-free constrained optimization problem, we achieve significant performance improvements. On ViT-L/14, our method reaches 92.6% performance, approaching the 93.5% achieved by multi-task learning—a substantial narrowing of the gap. We have revised the description of model merging requirements in Line 18, and greatly appreciate your suggestion.
> ___
> > Q2: Theoretically, it is also unclear why "only take gradient steps in the direction orthogonal to the shared space" (Line 68) can help us achieve this ambitious goal. The argument is not convincing.
>
> A2: The optimization objective in Eq. (5) promotes orthogonality between task vectors to mitigate conflicts, **while multi-task learning similarly emphasizes shared representations**. Parameters between similar tasks can be shared (e.g., applying the MNIST task vector improves accuracy on SVHN). Therefore, we propose constructing a shared subspace $S_{share}$ to preserve common representations. By constraining task vector optimization to reduce updates along $S_{share}$, we maintain shared knowledge while minimizing the gap for each task as defined in Eq. (5).
>
> Ablation studies demonstrate a 3.5% improvement with $S_{share}$. Table 7 presents a comparison of different gradient directions, revealing dataset-specific performance variations. Our method achieves significant improvements on the DTD dataset while showing decreased performance on SVHN. This pattern stems from DTD's reliance on rich textural features that are preserved in $S_{share}$. In contrast, SVHN's visual representations differ substantially from other tasks, making the primary components in $S_{share}$ less suitable. This observation is further validated by examining the performance gap between pre-trained and fine-tuned models: SVHN exhibits the lowest pre-trained performance (31.4%) but achieves remarkable results after fine-tuning (97.5%), indicating its strong dependence on task-specific features. In summary, our approach effectively preserves shared knowledge across tasks while achieving optimal overall performance.
> ___
> > Q3: The connection between the motivation (keeping task-specific information) and the method is not very clear.
>
> A3: To isolate task-specific information, the task vector is defined as $\tau_i = \theta_i - \theta_0$ to capture unique characteristics for each task. While preserving task-specific information through simple vector addition is straightforward, the challenge in model merging lies in managing conflicts between multiple tasks. This challenge becomes evident in Figure 1, which demonstrates how performance consistently declines across all merging methods as the number of tasks increases, directly reflecting increased task conflicts.
>
> As shown in Eq. (3), we measure the gap between the merged model and individual models **in terms of task-specific losses**. To alleviate conflicts, we introduce a modification vector $\Delta$ for each task vector. This leads to our optimization objective in Eq. (4), which aims to achieve optimal cross-task performance by optimizing $\Delta$. Through this optimization process, the merged model approximates the behavior of task-specific models while effectively resolving conflicts. In short, by minimizing our proposed loss function, we ensure the merged model preserves essential task-specific information.

---

> > ### Comment · Reviewer_tyLA · 2025-04-05
> >
> > I would like to thank the authors for providing a detailed rebuttal. However, my concerns mentioned above were not solved, including the performance gap and the theoretical advantage of the proposal. Therefore, I decided to maintain my original ratings.

---

> > > ### Author Response · Authors · 2025-04-05
> > >
> > > Thank you again for your thorough review. We incorporate your constructive suggestions to better explain our method. Regarding concerns about the performance gap, please refer to our recent discussion with Reviewer `QBR6`. We acknowledge that theoretical advantage is not our primary contribution. Our paper directly models the multi-task model merging problem and empirically validates our motivation through experimental evidence.

---

### Official Review · Reviewer_QBR6 · 2025-03-14

**Overall Recommendation:** 3

**Summary:**

The authors introduced an approach to merging tasks for a multi-task learning purpose while maintaining performance comparable to task-specific models. They formulated the problem as a constrained optimization task, solved using adaptive projected gradient descent. To facilitate task merging, they introduced a modification vector for each task, acting as a correction mechanism. To achieve this, they constructed a shared subspace using SVD to capture common features, optimizing within this space to minimize task conflicts. The gradient of the modification vector is decomposed into two components: one projected onto the shared subspace and the other orthogonal to it. Additionally, they introduced merging coefficients based on the norm of task vectors to mitigate the dominance of any single task’s gradient influence.

**Claims And Evidence:**

Yes

**Essential References Not Discussed:**

Yes, the Weight-Ensembling Mixture of Experts (WEMoE) method for multi-task model merging was introduced in the paper "Merging Multi-Task Models via Weight-Ensembling Mixture of Experts", published at ICML 2024. Additionally, an extended arxiv version, "Efficient and Effective Weight-Ensembling Mixture of Experts for Multi-Task Model Merging" (E-WEMoE), further refines this approach. Both papers should be included in the related work section for a comprehensive discussion.

**Ethical Review Concerns:**

Figure 3 closely resembles the one presented in the paper "Efficient and Effective Weight-Ensembling Mixture of Experts for Multi-Task Model Merging", sharing the same representation and color scheme. However, there is no proper citation to this arXiv paper. Notably, this arXiv paper is an extension of the ICML 2024 accepted paper, "Merging Multi-Task Models via Weight-Ensembling Mixture of Experts", and both papers report better results than the paper currently under review. Given that the authors reproduced a highly similar image without citing the original work—regardless of whether they are the same authors—the omission  appears intentional, particularly since the prior work (in both papers) demonstrates superior performance.

**Experimental Designs Or Analyses:**

No

**Methods And Evaluation Criteria:**

Yes

**Other Comments Or Suggestions:**

None

**Other Strengths And Weaknesses:**

Strengths:
They applied their approach on Vision and NLP tasks.
Weakness:
Each dataset should have a brief description.
Most of the included datasets focus on a single task, primarily classification. Can this approach be applied to heterogeneous MTL?
SVD is computationally expensive. Can this approach be applied to Llama 2 or Llama 3?
Traditional MTL needs to be clarified more. For instance, what is its architecture?
The results were not compared against the WEMoE published in this paper “Merging Multi-Task Models via Weight-Ensembling Mixture of Experts” and  E-WEMoE frameworks presented in the paper "Efficient and Effective Weight-Ensembling Mixture of Experts for Multi-Task Model Merging". Additionally, Figure 3 is similar to one in the paper "Efficient and Effective Weight-Ensembling Mixture of Experts for Multi-Task Model Merging”.
For vision tasks, your results fall short compared to those reported in the paper “Merging Multi-Task Models via Weight-Ensembling Mixture of Experts” and the paper "Efficient and Effective Weight-Ensembling Mixture of Experts for Multi-Task Model Merging".

**Questions For Authors:**

None

**Relation To Broader Scientific Literature:**

The authors are trying to tackle the issue with merging parameters of the model achieving good performance comparable to task-specific methods by reducing conflict of tasks. They mentioned most of the other literature work that addressed this issue.

**Theoretical Claims:**

No

---

> ### Author Rebuttal · Authors · 2025-03-30
>
> > Q1: The omission appears intentional, since the prior work demonstrates superior performance.
>
> A1: Our approach differs fundamentally from [1,2] in both setting and methodology. Our objective is to close the performance gap between model merging and multi-task learning **without introducing additional computation and memory requirements—a core and previously unresolved challenge in model merging research**.
>
> - ### Parameter
>   MoE architecture preserves MLP layers from each fine-tuned task-specific model and the pre-trained model, while additionally training a router module. In contrast, our merged model maintains a standard model size. The parameter comparison for ViT-B/32 (8 tasks) is as follows:
> ||Total Parameters|
> |-|:-:|
> |Individual|113.45M|
> |Ours|113.45M|
> |WEMoE|573.96M|
>
>   The primary motivation for model merging is parameter reduction. If performance were the sole consideration, retaining each task-specific model would be the trivial solution. **Our method aims to compress multiple models (whether 8 or even 20) into a single standard-sized model, which aligns with the typical settings in model merging and multi-task learning**. As MoE methods (89.4%) exceed the performance upper bound of multi-task learning (88.9%), comparing our approach directly with MoE would be inappropriate.
>
> - ### Data Requirements
>   MoE approaches employ unlabelled test datasets to train the router module, whereas our optimization of task vectors is data-free. The performance benefits from test-time adaptation are self-evident. **Merging based solely on model parameters is more practical and represents the focus of most model merging methods**.
>
> - ### Computational Overhead
>   Static merging maintains **inference costs** equivalent to standard models, while MoE dynamic merging consumes more memory and computational resources (router + activated experts $k$). The inference phase memory usage comparison is as follows:
> ||ViT-B/32 (8 tasks)|ViT-B/32 (20 tasks)|ViT-L/14 (8 tasks)|
> |:-:|:-:|:-:|:-:|
> |Ours|963.42MB|963.42MB|3772.63MB|
> |WEMoE|2750.65MB|5346.00MB|10063.64MB|
>
>   Similarly, test-time adaptation incurs additional **training costs**, while our method requires only lightweight training overhead (as shown in Table 10 of our paper):
>   ||ViT-B/32 (8 tasks)|ViT-L/14 (8 tasks)|ViT-B/32 (8 tasks)|ViT-L/14 (8 tasks)|
>   |:-:|:-:|:-:|:-:|:-:|
>   |Ours|729MB|2448MB|2.02min|5.18min|
>   |WEMoE|3744.19MB|24535.53MB|7.07min|56.84min|
>
>   Notably, our method can be trained layer by layer, enabling model merging for large models with minimal memory requirements.
>
> - ### Regarding Figure 3
>   Figure 3 visualizes task vector magnitudes, highlighting a phenomenon inherently observable across domain benchmarks. E-WEMoE and DOGE propose different approaches to address this phenomenon. Figure 3 was drawn with assistance from E-WEMoE authors to create a new version. **Associating performance gaps with non-citation introduces a conceptual misunderstanding, as fair comparison is impossible due to differing settings.** Meanwhile, we compare our approach with state-of-the-art methods in both data-free and test-time adaptation scenarios (described in lines 314-328). We appreciate your feedback and will introduce MoE-like methods and the clear differences.
> ___
> > Q2: Each dataset should have a description. Most of the included datasets focus on classification. Can this approach be applied to heterogeneous MTL?
>
> A2: We will add descriptions for each dataset. Model merging in CV indeed focus primarily on classification tasks, following common experimental settings (as acknowledged by Reviewer `97J9`). Research on heterogeneous model merging remains limited, with existing work mainly centered on VGG and ResNet architectures using CIFAR datasets. We would welcome suggestions for appropriate benchmarks to explore this direction.
> ___
> > Q3: SVD is computationally expensive. Can this approach be applied to Llama 2 or Llama 3?
>
> A3: SVD computation only needs to be performed once at the beginning. As shown in Table 10, which details the computation overhead, **our approach requires minimal memory and time**. We conducted experiments following standard LLM settings, completing the merging in 58 min on a single A100 GPU. We report normalized scores on merging WizardLM-13B (Instruction-Following), WizardMath-13B (Math), and llama-2-13b-code-alpaca (Code). Our method achieves optimal average performance across tasks.
> ||AlpacaEval|GSM8K|MATH|HumanEval|MBPP|Avg.|
> |:-:|:-:|:-:|:-:|:-:|:-:|:-:|
> |Individual|100.0|100.0|100.0|100.0|100.0|100.0|
> |TA|102.7|91.0|70.5|50.0|87.7|80.4|
> |TIES|98.1|97.4|68.1|60.0|89.4|82.6|
> |TA + DARE|103.1|88.0|72.5|63.3|92.9|84.0|
> |TIES + DARE|107.9|90.3|65.6|80.0|92.4|87.2|
> |Ours|107.5|105.0|94.4|56.7|86.5|**90.0**|
> ___
>
> [1] Merging Multi-Task Models via Weight-Ensembling Mixture of Experts. ICML 2024.
> [2] Efficient and Effective Weight-Ensembling Mixture of Experts for Multi-Task Model Merging. ArXiv 2024.

---

> > ### Comment · Reviewer_QBR6 · 2025-04-05
> >
> > 1. The current response is contradictory. Authors mentioned "If performance were the sole consideration, retaining each task-specific model would be the trivial solution", yet the paper’s primary comparison focuses only on accuracy. Furthermore, the authors do not provide any direct comparison regarding computation or memory efficiency against the state-of-the-art, which undermines their claim.
> >
> > Regarding the comparison to WEMoE, the authors argue that it is unfair to compare their approach against WEMoE ( MoE dynamic merging) due to differences in merging strategies. However, both methods fundamentally merge parameters, where WEMoE does so dynamically based on test data, while the proposed approach employs a static merging method. Given that both tackle the same problem using the same datasets, the comparison appears valid. Moreover, the prior work, WEMoE, was evaluated using the same benchmark, such as Adamerging (test adaptation), Ties-Merging, and Task Arithmetic (data-free methods), which were used by the authors in this paper to evaluate their approach.
> >
> > Therefore, the justification for claiming unfairness in comparison to WEMoE is unconvincing. The authors should explicitly include a discussion of these prior works in their manuscript, clearly outlining the pros and cons of their approach relative to them. In particular, while their method may offer improvements in computational and memory efficiency, it is important to address the fact that WEMoE surpasses their approach in accuracy. A balanced discussion of these trade-offs—accuracy versus resource efficiency—would provide a more comprehensive evaluation of the contributions of this work.
> >
> >
> >
> > 2. The computational overhead comparison supports the claim that the static merging approach offers significant savings compared to dynamic MoE methods (i.e., WEMoE) . However, the analysis would be stronger if it quantified these benefits—for example, by stating the percentage reduction in memory usage and computation overhead relative to WEMoE, and reporting any corresponding percentage loss in accuracy. Detailed discussion of the trade-offs between resource savings and potential accuracy impacts should be included as it would provide a more comprehensive evaluation of the method's overall effectiveness.
> >
> > Other comments:
> >
> > 3. For Figure 3, it appears similar to one presented in a previous paper. The authors stated explicitly that they created this version with assistance from the E-WEMoE authors, indicating that it is derived from prior work (including their code). Therefore, it is essential that they provide proper citation to the original source in the figure caption.
> >
> > 4. The authors did not respond to this question "Traditional MTL needs to be clarified more. For instance, what is its architecture?".

---

> > > ### Author Response · Authors · 2025-04-05
> > >
> > > Thanks for your time and feedback. Please find point-by-point responses to your concerns below:
> > > ___
> > > >Q1: The current response is contradictory, yet the paper’s primary comparison focuses only on accuracy.
> > >
> > > A1: Our response is not contradictory. The target of model merging is to merge multiple models into a single model that approaches the accuracy of task-specific models. Model merging has developed rapidly, leading to inconsistencies across many works. **This is a current issue in the field, as there is no clear distinction based on parameters, data requirements, and computational costs.**
> > > For example, SOTA dynamic merging methods like EMR merging and Twin merging, which function as lightweight WEMoE, also did not compare with WEMoE in their evaluations, instead comparing against AdaMerging (test adaptation) and Ties-Merging (data-free). As stated in the paper, **DOGE is a plug-and-play method**—we incorporat it into classic methods from both test adaptation (AdaMerging) and data-free (Task Arithmetic) categories, achieving SOTA performance in static merging. DOGE can similarly enhance dynamic methods by replacing their weighted averaging components.
> > > ___
> > > >Q2: However, the analysis would be stronger if it quantified these benefits—for example, by stating the percentage reduction in memory usage and computation overhead relative to WEMoE.
> > >
> > > A2: We will provide a comprehensive comparison table and include a detailed discussion of previous works in the manuscript, offering readers a thorough evaluation:
> > >
> > > | Method| Parameters | Router | Data | Parallel | Performance |
> > > |:-|:-:|:-|:-|:-:|:-:|
> > > | TA [1]| 1$\times$ | -  | -  | static  | 69.1|
> > > | AdaMerging [2]| 1$\times$  | -  | unlabeled test dataset  | static  | 80.1  |
> > > | TA+DOGE | 1$\times$  | -  | - | static   | 81.0 (**$\uparrow$ 11.6**)|
> > > | AdaMerging+DOGE| 1$\times$ | - | unlabeled test dataset  | static   | 85.9 (**$\uparrow$ 5.8**)  |
> > > | Surgery [3]| >1$\times$| -| unlabeled test dataset | static  | 80.9 |
> > > |--|--|--|--|--|--|
> > > | WEMoE [4]| 5$\times$  | trained router| unlabeled test dataset  | dynamic  | 89.4|
> > > | EMR merging [5]| 4$\times$| perfect router  | - | dynamic  | 88.7|
> > > | Twin merging [6]| 2.25$\times$| trained router| labeled validation dataset | dynamic | 86.1|
> > > |--|--|--|--|--|--|
> > > | Traditional MTL| 1$\times$ | - | -| - | 88.9|
> > > | Multiple Models| 8$\times$ | - | - | - | 90.8 |
> > >
> > > As shown, merging multiple models into a single model presents significant challenges. DOGE, as a plug-and-play method, substantially improves accuracy. **Dynamic merging face parallelization issues during inference**, requiring either dynamic I/O loading of task-specific modules or storing all modules in GPU memory. EMR merging needs priors during inference to load corresponding modules, while WEMoE and Twin merging train routers to select modules. We believe methods should be classified before conducting fair comparisons within each category. **Otherwise, according to the no free lunch theorem, MoE methods will always outperform any static merging methods simply due to their larger parameter count.**
> > > ___
> > > >Q3: It is essential that they provide proper citation to the original source in the figure caption.
> > >
> > > A3: Thank you for bringing this oversight. We will provide proper citation in the figure caption.
> > > ___
> > > >Q4: Traditional MTL needs to be clarified more. For instance, what is its architecture?
> > >
> > > A4: We apologize for the previous omission. As explained in Appendix C (Lines 582-583), Traditional MTL trains a single base model on all tasks simultaneously. The architecture is the standard base model.
> > > ___
> > > To summarize our contribution again: We frame model merging as a constrained optimization problem, propose projective gradient descent that optimizes a data-free objective, and design task-aware merging coefficients. Comprehensive experiments validate our plug-and-play capability.
> > >
> > > Your discussion regarding MoE methods has helped us provide a more comprehensive evaluation in our paper. **We believe that clearer categorization and comparison will benefit the model merging community as a whole.** Thank you sincerely, and we wish you a pleasant day.
> > > ___
> > > [1] Editing Models with Task Arithmetic. ICLR 2023.
> > > [2] AdaMerging: Adaptive Model Merging for Multi-Task Learning. ICLR 2024.
> > > [3] Representation Surgery for Multi-Task Model Merging. ICML 2024.
> > > [4] Merging Multi-Task Models via Weight-Ensembling Mixture of Experts. ICML 2024.
> > > [5] EMR-Merging: Tuning-Free High-Performance Model Merging. NeurIPS 2024.
> > > [6] Twin-Merging: Dynamic Integration of Modular Expertise in Model Merging. NeurIPS 2024.

---

### Official Review · Reviewer_97J9 · 2025-03-14

**Overall Recommendation:** 3

**Summary:**

This paper addresses the challenge of merging multiple task-specific models into a unified model without accessing their original training data. The authors identify critical limitations in existing methods, such as discarding task-specific information during conflict resolution and over-enforcing orthogonality, which erodes shared knowledge. They propose DOGE, a constrained optimization framework that minimizes performance gaps via data-free gradient descent, projects updates orthogonally to a shared subspace (preserving common representations), and employs task-aware merging coefficients derived from task vector norms.

**Claims And Evidence:**

- Correct me if wrong, to use the Taylor expansion, the expanded point and the pretrained model should be very close. May need to point this out and justify.
- In addition, did the author evaluate the performance of using this 1st order Taylor expansion to approximate the loss to validate this choice?

**Essential References Not Discussed:**

For the Taylor expansion part, it would be helpful to cite a related work (MAP: https://arxiv.org/pdf/2406.07529) which also uses Taylor expansion to approximate the loss function / evaluation metric.

**Experimental Designs Or Analyses:**

Yes, I checked the experiments. As I mentioned before, it would be great to add comparisons between DOGE with other strong baseline methods such as EMR merging and Twin merging.

**Methods And Evaluation Criteria:**

Methods:
- In algorihtm 1, the authors should define $\Delta$, whether it is input or how it is initialized.

Evaluation:
- The benchmark datasets are commonly used in task vector based model merging.
- However, I would like to see the comparison between DOGE with other strong baseline methods such as EMR merging, Twin merging.

**Other Comments Or Suggestions:**

- In methodology section, $\lVert\cdot \lVert_{Gap}$ and $\lVert\cdot \lVert_{Sshare}$ make it seems like you are defining some new norms. I would avoid using them as subscripts of the norm symbol.
- Table 5 and 6 are not numbered in the order they appeared in the paper.
- Table 5: it is interesting the tasks selected MNIST and EuroSAT are the relatively easier tasks to the ViT models. It would be interesting to see the generalization performance on SUN397, DTD, and Cars.

**Other Strengths And Weaknesses:**

Strengths:
- Empirical results (performance gains, robustness to task scaling, cross-domain generalization) convincingly demonstrate DOGE’s practical efficacy.

- The plug-and-play design and compatibility with architectures like ViT/LoRA are validated experimentally.

Weaknesses:
- Since the method requires additional optimization and additional modification vectors for each task, I would like the authors to present the additional time/space that DOGE requires.

**Questions For Authors:**

- Is $\Delta$ a task-specific vector? Since you mentioned $\Delta$ is a modification vector to each task vector, and it is not indexed by the task, it was a bit confusing to me at the beginning. Maybe rephrase it to "a universal modification vector to each task vector".

**Relation To Broader Scientific Literature:**

This paper is related to prior ideas including twin-merging (modulating shared and exclusive knowledge) and representation surgery (trying to make the representation of the merged model close to each individual task). It uses Taylor expansion to approximate the loss without using any data (similar to the idea in MAP (using Taylor expansion to approximate loss function)).

**Theoretical Claims:**

There is no theoretical claims in this paper.

---

> ### Author Rebuttal · Authors · 2025-03-31
>
> > Q1: Taylor expansion may need to point this out and justify.
>
> A1: During fine-tuning, parameter evolution in pre-trained models is frequently minimal, indicating that training remains within the tangent space where Taylor expansion closely approximates network behavior. This aligns with MAP, which examines task vector magnitudes and employs 2nd Taylor expansion to approximate metrics. It provides a formal proof regarding the negligibility of the remainder in Taylor series, and interestingly proposes using linear regression to estimate Hessian. Thanks for suggesting this related work! It strengthens our theoretical foundation, and we include this reference to further substantiate the rationale behind our approach.
>
> We examined the difference between the 1st order Taylor expansion and the original loss, finding them to be within the same order of magnitude, confirming the accuracy of the estimation. Since calculating the gradient $\nabla_{\theta}\mathcal{L}_j(\theta_0)$ requires specific data $\mathcal{D}_j$, we used task vector $\tau_j$ as an approximation. Interestingly, when we attempted to optimize using actual gradients computed from specific data, we observed performance degradation. We attribute this to highly unstable gradients at the initialization, which complicated the optimization process. Thus, approximating the original loss using task vectors appears to be the superior way.
> ___
> > Q2: Whether $\Delta$ is input or how it is initialized.
>
> A2: $\Delta$ is initialized as a zero tensor with the same shape as the task vector.
> ___
> > Q3: Strong baseline methods such as EMR merging, Twin merging.
>
> A3: As discussed in Twin merging [2], they all belong to dynamic merging, which requires **additional storage for task-specific modules**. Such methods face **parallelization** challenges during inference, necessitating either dynamic I/O loading of task-specific modules or storing all modules in GPU memory. EMR merging requires priors during inference to load corresponding modules, while Twin merging trains a router using validation datasets to select modules.
>
> Both EMR merging and Twin merging can be viewed as lightweight WEMoE [3], yet they still impose storage demands (**2.25× our approach**). For instance, EMR merging's proposed mask implementation still uses 8-bit Bool types, and Twin merging's module reconstruction $U\Sigma V$ requires matrix operations that may not reduce peak GPU consumption. Notably, these approaches avoid direct comparison with WEMoE, which is unsurprising. According to the no free lunch theorem, performance increases with the number of retained parameters, with complete task-specific models representing the upper performance bound.
>
> Our approach, by contrast, is a static merging plug-and-play method (like TA and Ties merging) that maintains standard model size and enables parallelized inference. We compare our method with SOTA static merging approaches such as AdaMerging and PCB-Merging. **We believe methods should first be classified before conducting fair comparisons within each category. Otherwise, MoE methods will always outperform others simply due to larger parameter count.**
> ___
> > Q4: Present the time/space that DOGE requires.
>
> A4: We have reported training time and memory usage in Table 10 of the Appendix, demonstrating remarkably efficient performance with only 121 seconds total training time and a memory usage of 729MB. We will relocate this information to the main text.
> ___
> > Q5: Generalization performance on SUN397, DTD, and Cars.
>
> A5: Based on your request, we conducted experiments evaluating generalization on three unseen tasks when merging five other tasks. The results reveal that SUN397, DTD, and Cars datasets pose challenges for ViT models, while MNIST/EuroSAT show limited generalization to these complex tasks. Despite this, our method consistently outperformed other model merging approaches by a significant margin.
> |Method|Seen||||||Unseen||||
> |:-:|:-:|:-:|:-:|:-:|:-:|:-:|:-:|:-:|:-:|:-:|
> ||RESISC45|SVHN|GTSRB|MNIST|EuroSAT|Avg.|SUN397|Cars|DTD|Avg.|
> |Pre-trained|60.6|23.5|30.4|47.6|45.6|41.5|63.2|59.9|43.9|55.6|
> |Task Arithmetic|52.8|83.9|71.1|97.7|61.9|73.5|27.9|25.0|26.4|26.4|
> |Ties-Merging|74.6|89.1|81.8|97.7|73.7|83.4|57.5|51.9|38.7|49.4|
> |AdaMerging|73.5|76.0|81.5|97.4|69.4|79.6|42.3|37.8|32.0|37.4|
> |DOGE TA|82.6|89.4|89.0|98.6|92.3|**90.4**|58.7|54.3|41.4|**51.5**|
> ___
> > Q6: Is $\Delta$ a task-specific vector?
>
> A6: Thanks for the suggestion. $\Delta$ is a universal modification vector to each task vector. In our experiments, using a universal modification vector yields performance nearly identical to that of task-specific modification vectors, as they are mathematically equivalent when optimizing Eq.(5).
> ___
> [1] EMR-Merging: Tuning-Free High-Performance Model Merging. NeurIPS 2024.
> [2] Twin-Merging: Dynamic Integration of Modular Expertise in Model Merging. NeurIPS 2024.
> [3] Merging Multi-Task Models via Weight-Ensembling Mixture of Experts. ICML 2024.

---

### Official Review · Reviewer_B1Xi · 2025-03-16

**Overall Recommendation:** 3

**Summary:**

This paper proposes a new perspective on model merging—treating it as a multi-task learning problem rather than merely a parameter-level combination of multiple expert models. The main idea is to preserve each task’s strong performance while reconciling the potential conflicts that arise when unifying several task-specific models into a single, merged network. To that end, the authors introduce an approach they call “Adaptive Projective Gradient Descent” (DOGE). This method formulates the model-merging goal as a constrained optimization problem that minimizes the “performance gap” between the merged model and each of the individual expert models, while explicitly retaining cross-task shared representations.

The procedure has three core steps. First, it refines (or “modifies”) each task-specific vector so that merging doesn’t simply discard conflict-ridden parameters that might actually be performance-critical. Second, it projects the gradients of these task modifications onto a shared subspace to maintain overlapping knowledge across tasks rather than forcing all task vectors into near-orthogonality. Third, it adapts the merging coefficients in a “training-free” way that is reminiscent of how adaptive optimizers dynamically adjust the learning rate; effectively, the magnitude of each merging coefficient is scaled inversely by the norm of the corresponding task vector. The overall pipeline is then shown to achieve strong performance across diverse architectures (vision and language) and tasks (classification, generation) without requiring access to the original datasets.

**Claims And Evidence:**

- Improved performance on merged models: DOGE is claimed to achieve higher accuracy than previous data-free merging approaches by better preserving task-specific information while retaining shared representations.
- Effectiveness of gradient projection: By projecting gradient updates orthogonally to the shared subspace, the method aims to resolve task conflicts without sacrificing common knowledge.
- Task-aware coefficient design: The adaptive (training-free) merging coefficients based on task vector norms provide a natural way to balance gradient contributions across tasks.
The evidence supporting these claims comes from extensive experimental results on multiple benchmarks in both vision and NLP domains. - Quantitative comparisons across a variety of baselines (non-merging methods, data-free approaches, and test-time adaptation techniques) and detailed ablation studies show significant improvements. Although the experimental evidence is robust, one might wish for more discussion regarding statistical variability (e.g., more error bar analysis or significance testing) to further solidify the claims.

**Essential References Not Discussed:**

Some relevant work that tackle model merging on subspace need to be discussed:
- Gargiulo, A. A., Crisostomi, D., Bucarelli, M. S., Scardapane, S., Silvestri, F., and Rodola, E. Task singular ` vectors: Reducing task interference in model merging. arXiv preprint arXiv: 2412.00081, 2024.
- Stoica, G., Ramesh, P., Ecsedi, B., Choshen, L., and Hoffman, J. Model merging with svd to tie the knots. arXiv preprint arXiv: 2410.19735, 2024.

**Experimental Designs Or Analyses:**

The experimental setup is comprehensive:

- The authors evaluate on eight-task vision benchmarks using CLIP-based ViT-B/32 and ViT-L/14 models, as well as on eight-task language benchmarks using LoRA fine-tuned Flan-T5 models.
- Multiple baselines, including both data-free and test-time adaptation methods, are used for comparison.
- Detailed ablations assess the contribution of each module (∆ optimization, shared subspace projection, and adaptive λ), lending credibility to the claims about each component’s effectiveness.
- Additional experiments on unseen tasks and corrupted test sets reinforce the method’s robustness.

One minor suggestion is to include more explicit details on computational overhead (I found some recent work also update ∆ by gradient descent but not sure how expensive this procedure is) and convergence behavior across varying numbers of tasks.

**Methods And Evaluation Criteria:**

The proposed method is well-motivated and methodologically sound. Key components include:

- Data-Free Objective: Derived via a first-order Taylor expansion, the objective minimizes the loss gap between the merged model and each individual model by approximating the unavailable gradient with the task vector.
- Shared Subspace Construction: SVD is used to extract task-specific subspaces, which are then combined and refined to form a shared subspace that guides the gradient projection.
- Adaptive Merging Coefficients: Interpreting task vectors as cumulative gradients leads to a natural formulation where merging coefficients play a role akin to adaptive learning rates.
- The evaluation criteria—such as average accuracy across tasks (or Spearman’s ρ for STSB in NLP) and performance on out-of-distribution or unseen tasks—are appropriate for demonstrating the effectiveness and generalization of the method. The comprehensive experiments across different architectures and task modalities further validate the method’s practicality.

**Other Comments Or Suggestions:**

- Clarity in Derivations: Some steps in the derivation of the data-free objective could be elaborated further. A step-by-step explanation with more intuition would improve readability. For example, ∥θ∗ − θi∥_Gap apprears in Eq 2 without introduction.
- Limitations and Future Work: It would be beneficial for the authors to include a discussion on potential limitations (e.g., cases where tasks are highly heterogeneous) and directions for future research.

**Other Strengths And Weaknesses:**

Strengths:

- Comprehensive Evaluation: The extensive experimental validation across both vision and language domains, along with detailed ablation studies, convincingly demonstrates the method’s effectiveness.
- Practical Relevance: The approach is designed to work in data-free scenarios, which is particularly appealing in settings where access to original training data is restricted due to privacy or logistical concerns.

Weaknesses:

- Theoretical Rigor: Some derivations, particularly the gradient approximations and the rationale behind using −τ_j as a proxy for the gradient, could benefit from a more rigorous treatment.
- Hyperparameter Sensitivity: The method involves several hyperparameters (e.g., the subspace basis size and global scaling factor η) whose selection may critically affect performance. More discussion on sensitivity analysis would be helpful.
- Computational Overhead: A deeper analysis of the additional computational costs (e.g., due to SVD and projection operations) would enhance understanding of the method’s scalability.

**Questions For Authors:**

- How sensitive is the overall performance to the choice of the subspace basis size and the global scaling factor η?

- Could you provide further empirical or theoretical justification for approximating ∇θLj(θ0) by −τ_j? Under what conditions might this approximation break down? Should I expect gradient-based MTL methods (https://github.com/thuml/awesome-multi-task-learning) can outperform task arithmetic (e.g. MGDA, CAGRAD, PCGRAD, IMTL...)

**Relation To Broader Scientific Literature:**

The paper is well-situated within the broader context of multi-task learning and model merging:

- It builds upon previous work in data-free model merging (e.g., Task Arithmetic, Ties-Merging) and test-time adaptation (e.g., AdaMerging).
- It draws connections to multi-task learning strategies that emphasize gradient alignment and modular architectures.

**Theoretical Claims:**

The paper provides heuristic derivations rather than fully formal proofs. Notably:

- The use of a first-order Taylor expansion to derive a data-free objective is a reasonable approximation given the unavailability of task data.
- Approximating the gradient of the task loss at the pre-trained model using the task vector (i.e., −τ_j) is intuitively justified by interpreting the task vector as an accumulation of gradients.
- The decomposition of the gradient into components within and orthogonal to the shared subspace is well-motivated, though the justification remains somewhat heuristic.

While these theoretical insights are plausible and backed by empirical evidence, a more rigorous treatment or further formal analysis would help strengthen the theoretical claims.

---

> ### Author Rebuttal · Authors · 2025-03-31
>
> Thanks for your detailed comments. We hope the following discussion can address your concerns!
> ___
> >Q1: Some relevant work that tackle model merging on subspace need to be discussed.
>
> A1: Thanks for suggesting additional relevant work. We will discuss them in related work: *TSV [1] aggregates task vectors within their subspaces via low-rank approximation and whitens matrices to minimize interference. KnOTS [2] aligns representation spaces between LoRA models using SVD, enabling the application of merging methods. Both these methods and ours recognize parameter low-rankness and implement merging within subspaces.*
> ___
> > Q2: Theoretical Rigor: The rationale behind using $−τ_j$ as a proxy for the gradient, could benefit from a more rigorous treatment.
>
> A2: Under the Neural Tangent Kernel assumption (i.e., fine-tuning often occurs in a linear regime), which has been validated in prior work [3,4], $\nabla_ {\theta}\mathcal{L}_ j(\theta_ 0)$ can be estimated as ${k\tau_ i}$ where $k < 0$. Here, $\tau_ j = θ_ T - θ_ 0 = -\sum_ {t=1}^T\alpha_ t\nabla_ {\theta_ t}\mathcal{L}_ j(\theta_t)$, where $\alpha_t$ represents the learning rate and $T$ denotes the update iteration. Given the linearity of parameters $θ_0$ in the vicinity, we have $\nabla_ {\theta_t}\mathcal{L}_ j(\theta_t) = \nabla_ {\theta_0}\mathcal{L}_ j(\theta_0)$. Therefore, we derive $\nabla_ {\theta}\mathcal{L}_ j(\theta_0)=-\frac{τ_j}{\sum_{t=1}^T\alpha_t}$.
> ___
> > Q3: Hyperparameter Sensitivity: The method involves several hyperparameters (e.g., the subspace basis size $k$ and global scaling factor $\eta$) whose selection may critically affect performance.
>
> A3: We have conducted experiments on the subspace basis size $k$ in Figure 4, which displays performance with varying rank ratios alongside the explained standard deviation. We also investigated the relationship between different projection directions and basis sizes. Additional sensitivity analysis for the global scaling factor $\eta$ is supplemented as follows:
> | η |0.01|0.02|0.03|0.04|0.05|0.06|0.07|0.08|0.09|
> |:-:|:-:|:-:|:-:|:-:|:-:|:-:|:-:|:-:|:-:|
> |ViT-B/32| 79.5 | 80.3 | 80.6 | 80.9 | **81.0** | 80.8 | 80.7 | 80.2 | 79.8 |
>
> The evaluation of  across values from 0.01 to 0.09 demonstrates that performance remains stable and even achieves higher results. (We did not conduct a specialized grid search, this setting was chosen because the calculated $\lambda$ was close to 0.3). This consistency across different $\eta$ values verifies the robustness of our approach and highlights the practicality of applying task-aware coefficients.
> ___
> > Q4: Computational Overhead: A deeper analysis of the additional computational costs (e.g., due to SVD and projection operations) would enhance understanding of the method’s scalability.
>
> A4: We have reported training time and memory usage in Table 10 of the Appendix, showing an efficient total training time of only 121 seconds and memory usage of 729MB. The SVD operation only needs to be executed once at the beginning, with a computational complexity of $O(\min(mn^2, m^2n))$. We appreciate your reminder and will relocate this to the main text. Moreover, we supplement the final version with convergence loss curves for 8 and 20 tasks, showing that convergence is typically achieved within 100 to 200 iterations.
> ___
> > Q5: Clarity in Derivations: A step-by-step explanation with more intuition would improve readability. For example, ∥θ∗ − θi∥_Gap apprears in Eq. (2) without introduction.
>
> A5: We apologize for any confusion caused. Eq. (2) is a brief mathematical summary presented before the detailed methodology. We have revised it and explained each symbol's meaning. Combined with the proof presented above, this will enhance the overall clarity of the derivation.
> ___
> > Q6: It would be beneficial for the authors to include a discussion on potential limitations and directions for future research.
>
> A6: Thanks for your suggestion. A potential limitation is the lack of consideration for heterogeneous model merging, which requires transformation when task vectors have inconsistent shapes or layer numbers. Regarding future research, we are extending our work to LLMs by merging WizardLM-13B, WizardMath-13B, and llama-2-13b-code-alpaca, achieving SOTA performance. For detailed table results, please refer to our response to Reviewer `QBR6`.
> ___
> > Q7: Should I expect gradient-based MTL methods can outperform task arithmetic.
>
> A7: Yes. Task arithmetic implements MTL in a training-free manner and can be viewed as a post-transfer approach for existing models, while MTL methods typically serve as performance upper bounds that we aim to approach.
> ___
> [1] Task Singular Vectors: Reducing Task Interference in Model Merging. CVPR 2025.
> [2] Model Merging with SVD to Tie the KnOTS. ICLR 2025.
> [3] Task Arithmetic in the Tangent Space: Improved Editing of Pre-Trained Models. NeurIPS 2023.
> [4] A Linearized Framework and A New Benchmark for Model Selection for Fine-Tuning. ArXiv 2021.

---

### Decision · Program_Chairs · 2025-05-01

**Decision:**

Accept (poster)

**Comment:**

This paper initially received borderline scores. The main concerns raised by the reviewers included: (1) lack of discussion on related work, such as subspace-based methods for model merging; (2) insufficient justification for the theoretical analysis, particularly the linearity assumption of the loss landscape near fine-tuned parameters; (3) limited analysis of computational cost; and (4) missing comparisons with MoE-based methods, such as approaches using routers.

During the rebuttal, the authors addressed most of these concerns effectively, leading to a consensus among reviewers in favor of acceptance. In particular, the AC believes that comparing MoE-based and static model merging approaches purely in terms of accuracy is not very meaningful and that computational cost should be considered as part of the evaluation. For example, MoE approaches with a zero-compression rate are effectively equivalent to storing all fine-tuned parameters separately, diminishing their practical advantage.